



# Crustal structure of the East-African Limpopo Margin, a strike-slip rifted corridor along the continental Mozambique Coastal Plain and North-Natal Valley

Mikael Evain[1], Philippe Schnürle[1], Angélique Leprêtre[1,2], Fanny Verrier[1], Louise Watremez[3], Joseph Offei Thompson[1], Philippe de Clarens[4], Daniel Aslanian[1], and Maryline Moulin[1]

[1]IFREMER, Geosciences Marines, REM/GM/LGS, Centre de Brest, 29280 Plouzané, France
[2]LGO, IUEM, Place Nicolas Copernic, 29280 Plouzané, France
[3]Univ. Lille, CNRS, Univ. Littoral Côte d'Opale, UMR 8187 – LOG – Laboratoire d'Océanologie et de Géosciences, F-59000 Lille, France
[4]TOTAL, R&D, avenue Larribau, 64000 Pau, France
**Correspondence:** Mikael Evain (mikael.evain@ifremer.fr)

**Abstract.** Deep seismic acquisitions and a new kinematic study recently highlighted the presence of continental crust in both the southern Mozambique's Coastal Plain (MCP) and further offshore in the North Natal Valley (NNV). Such findings falsify previous geodynamic scenarios based on the kinematic overlap between Antarctica and Africa plates, thus profoundly impacting our understanding East-Gondwana break-up. Using an updated position of Antarctica with respect to Africa this study

reconsider the formation mechanism of East-African margins and most specifically of the Limpopo margin (LM). Coincident wide-angle and multi-channel seismic data acquired within the PAMELA project are processed to image the sedimentary and deep crustal structure along a profile that runs from the northeastern NNV to the Mozambique basin (MB) striking through the LM. This dataset is combined with companion deep seismic profiles and industrial onshore-offshore seismic lines to provide a robust scenario for the formation and evolution of the LM. Our P-wave velocity model consists of an upper sedimentary

sequence of weakly compacted sediments including intrusions and lava flows in the NNV while contourites and mass transport deposits dominates the eastern edge of the LM. This sequence covers a thick acoustic basement that terminates as a prominent basement high just west of the contourites and mass transport deposits domain. The acoustic basement has a seismic facies and velocity signature typical of a volcano-sedimentary basin and appears widespread over our study area extending toward the eastern MCP and NNV. Based on industrial well logs that calibrate our tectono-stratigraphic analysis we constrain its age to

be pre-Neocomian. We further infer that either strike-slip or trans-tensional deformation occurred at the basement high which sustained uplift up to the Neocomian. At depth, the crystalline basement and uppermost mantle velocity structures show a progressive eastward crustal thinning of continental crust along the edge of the MCP/NNV and up to the location of the basement high. On its eastern side, however, a corridor of anomalous crust depicts the velocity signature of a volcanic basement overlying lower continental crust. We infer that strike-slip rifting along the LM was accommodated at depth by ductile shearing respon-

sible for the thinning of the continental crust and an oceanward flow of lower crustal material. This process was accompanied by intense magmatism that extruded to form the volcanic basement and gave to the corridor its peculiar structure and mixed nature. The whole region remained at a relative high level and a shallow marine environment dominated during this period.





Only after break-up in the MB decoupling occurred between the MCP/NNV and the corridor allowing for the latter to subside
and being covered by deep marine sediments. We provide new insights into the early evolution and formation of the LM that
takes into account both kinematic and geological constraints. This scenario favors strike-slip rifting along the LM meaning that
no changes in extensional direction occurred between the rifting and the opening of the MB.

# 1   Introduction

East-Gondwana break-up led to the separation of four independent continental blocks, namely Africa, Antarctica-Australia,
Madagascar-India and Patagonia. This fragmentation strongly segmented the East-African margins in a succession of divergent
and strike-slip segments. To the north, southward motion of the Madagascar-India plate along the Davie Fracture Zone (DFZ)
opened the Somali Basin (SB). Similarly, the Antarctica-Australia plate drifted southward with respect to Africa along the
Mozambique Fracture Zone (MFZ) and opened the Mozambique Basin (MB). Finally, further south-west, the Patagonia plate
escaped from the coast of South-Africa along the Agulhas-Falklands Fracture Zone (AFFZ).

The Mozambique Coastal Plain (MCP) in southern Mozambique and its offshore prolongation the North Natal Valley (NNV)
form a wide buffer zone inherited from this complex break-up (Figure 1). It is surrounded to the north and east by continental
flood basalts (CFB) and associated dike swarms of Lower-Jurassic Karoo age (e.g., Jourdan et al., 2005; Watkeys, 2002), to
the east by the MB oceanic crust of Upper-Jurassic age (e.g., Mueller and Jokat, 2019) and to the south by the South Natal
Valley (SNV) oceanic crust that formed during Lower-Cretaceous while the Patagonia plate drifted (Goodlad et al., 1982). The
MCP-NNV area is known from industrial wells to present a thick sedimentary cover spanning from late Jurassic to Neogene.
It is, however, supposed to be floored by Karoo volcanics of the Stormberg series as solid datations are lacking (Flores, 1973;
Salman and Abdula, 1995). The oldest reliable age for basalts in the area concerns the Lower-Cretaceous Movene igneous that
disconformably cover the Karoo serie to the east of the Lebombos monocline (Flores, 1973; Watkeys, 2002).

Until recently and in the absence of deep seismic investigation, the crustal nature and geological history of the area have
remained speculative. Many have interpreted the presence of thickened oceanic crust and/or thinned continental crust with high
magmatic content based on potential field data, plate reconstruction and/or geological correlation with the conjugate margin in
Antarctica's Western Dronning Maud Land (DML) (e.g., Klausen, 2009; Leinweber and Jokat, 2011; Tikku et al., 2002; Watts,
2001). Consequently plate kinematic framework including so called 'tight' fit reconstruction have been privileged, where the
DML overlaps fully or partly the MCP/NNV, and an initial phase of rifting oblique or normal to the subsequent southward
plate drift is inferred (e.g., Cox, 1992; Klausen, 2009; Mueller and Jokat, 2019).

Today, this view is challenged by the recent acquisition of wide-angle seismic data which unravel the presence of a thick
and highly intruded continental basement over the entire MCP/NNV complex (Moulin et al., 2020). Conversely this confirms
previous works that interpreted continental crust in the area (Domingues et al., 2016; Hanyu et al., 2017) and brings strong





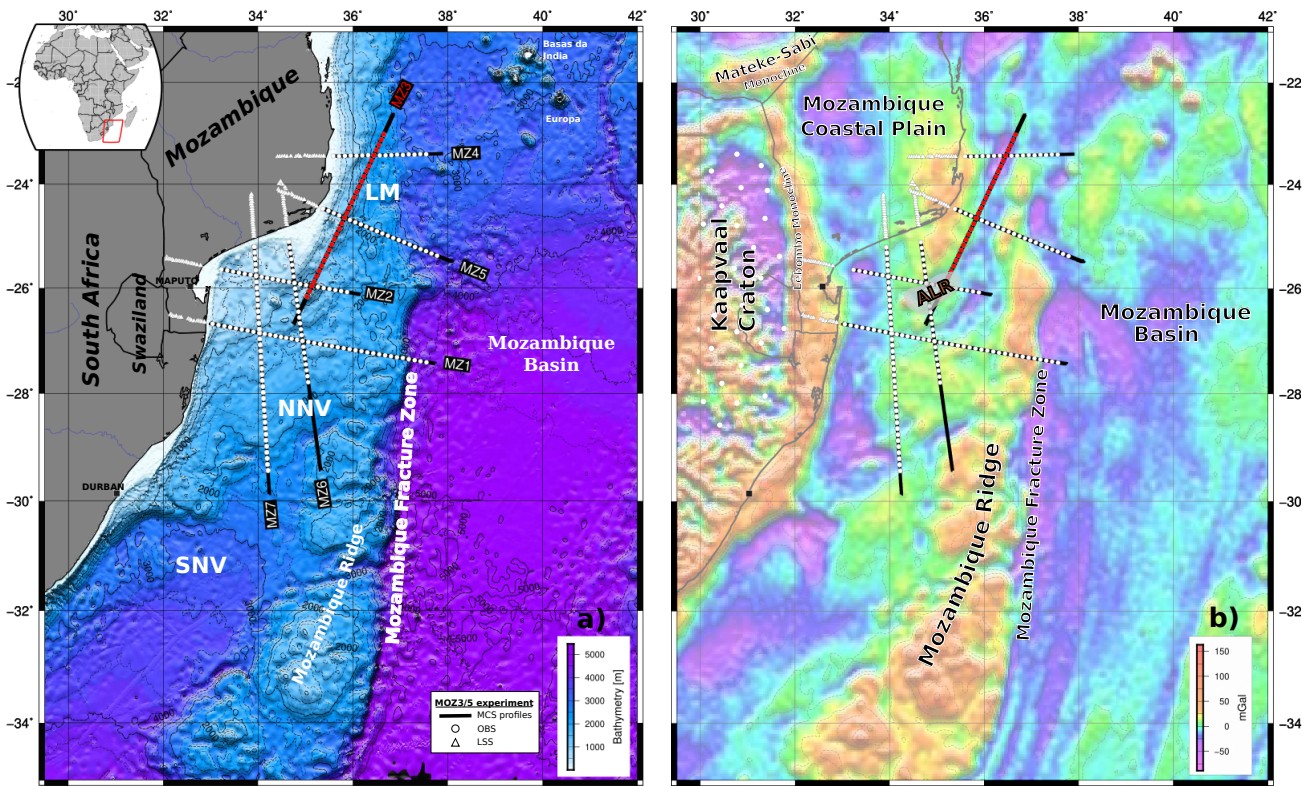

**Figure 1.** Map of the PAMELA MOZ35 seismic experiment. Combined MCS and wide-angle profiles are in black lines. MZ3 is highlighted in a red with red dots marking positions of OBS. Instruments (OBS and Landstations) and names of other profiles are in white. a) Bathymetric (GEBCO) grid. b) Free-air gravity anomaly grid (Pavlis et al., 2012; Sandwell et al., 2014) with identified magnetic anomalies and fracture zones (Mueller and Jokat, 2019). LM: Limpopo margin; NNV: North Natal Valley, SNV: South Natal Valley, ALR: Almirante Leite Ridge.

support for a 'loose' fit plate reconstruction where no plate overlap is imposed (Thompson et al., 2019). Furthermore, within

such framework, both rifting and subsequent spreading are inferred to proceed from a continuous N-S movement of Antarctica-Australia and Madagascar-India plates with respect to Africa with major implication on the dynamics of rifting along the East-African margins and the age of Gondwana break-up.

To shed new lights on the East-African margins structure and their formation mechanism the PAMELA (PAssive Margins Exploration Laboratories) project was initiated in the early 2010th by TOTAL and IFREMER in collaboration with French

universities (Université de Bretagne Occidentale, Université Rennes 1, Université Pierre and Marie Curie), the CNRS and the IFPEN. This study presents a combined multi-channels and wide-angle seismic profile acquired during the 2016 PAMELA





MOZ3-5 cruise (Moulin and Aslanian, 2016; Moulin and Evain, 2016) jointly with six other similar profiles presented in companion papers (Leprêtre et al., b, a; Verrier et al.; Schnürle et al.; Watremez et al.). We produced a p-wave velocity model of the sedimentary and deep crustal structure along this line that runs from the NNV toward the corner of the MB located east

of the MCP (Figure 1) and interpret it jointly with a tectono-stratigraphic analysis of the area to reveal the peculiarities of the strike-slip Limpopo Margin (LM) and infer its formation and evolution processes.

## 2 Geological setting

East Gondwana break-up initiated on a heterogeneous lithosphere that was last reworked and assembled during the PanAfrican Orogeny about 720 to 550 Ma (Guiraud et al., 2005; Jacobs and Thomas, 2004). In southern Africa, PanAfrican structures

include the Zambezi, Lurio and Mozambique belts. They linked together Archean and Proterozoic cratons (Zimbabwe and Kaapvaal) and belts (Limpopo, Namaqua-Natal) to their equivalent in Antarctica's Western Maud Land, namely the Grune-hogna Craton and Maud Belt (Bingen et al., 2009; Jacobs et al., 2008; Riedel et al., 2013).

The Karoo sedimentary sequence and subsequent Karoo magmatism attest of widespread intercontinental tectonic and mag-matic activities before Jurassic break-up. Karoo sediments consists of Late-Paleozoic to Early-Jurassic deposits within marginal

basins in South Africa (Catuneanu, 2004) and interconnected rifts in Antarctica, Eastern Madagascar, and Central and East Africa (Daly et al., 1989; Elliot and Fleming, 2004; Geiger et al., 2004; Salman and Abdula, 1995). They were largely covered by Karoo continental flood basalts between 185 and 177 Ma while, up to 172 Ma, sills and dykes also massively intruded Precambrian basement structures in southern Africa and Antarctica's Dronning Maud Land (Cox, 1992; Hastie et al., 2014; Jourdan et al., 2007).

Karoo sediments and volcanics of the Lebombo-Mateke-Sabi monoclines outline the northern and western boundaries of the MCP/NNV (Figure 1). To the east, the MFZ signs the southwards motion of Antractica-Australia plate that led to the opening of the MB in late Jurassic. Generally dated around 155 Ma on both the Mozambique margin and its conjugate Riiser-Larsen Sea in Antarctica (Jokat et al., 2003; König and Jokat, 2010; Thompson et al., 2019) other studies consider the oldest magnetic anomalies found on the northeastern corner of the MB to be  166-164 Ma (Leinweber and Jokat, 2012; Mueller and Jokat,

2019). The situation slightly differ on the opposite corner of the MB where early rifting concentrated in the offshore Zambezi depression before a rift jump isolated the Beira continental block from the rest of margin (Mahanjane, 2012; Mueller et al., 2016). Just south of this Beira High (BH) the oldest magnetic anomaly identified is 157 Ma (Mueller and Jokat, 2019).

South of the NNV, the Ariel Graben (AG) is a strong amplitude negative gravity anomaly oriented SW-NE that marks a boundary with the Mozambique Ridge (MR). Leprêtre et al. (a) and (Moulin et al., 2020) have shown that the MR does

not extend further north but instead positive gravimetric anomalies there reflect the presence of sedimentary features such as contourites as commonly observed in the area (Thiéblemont et al., 2019). South of the AG, seismic and gravimetric data concord on a oceanic origin for the MR. Its thick basement and morphological characteristics suggest numerous episodes of volcanism emplaced on a triple junction in Lower-Cretacous (Fischer et al., 2017; Gohl et al., 2011; König and Jokat, 2010; Leinweber and Jokat, 2012).





Several dredges made along the steep eastern border of the MR facing the MFZ and on its southwestern edge facing the SNV, however, recovered Archean fragments of continental rocks and methamorphic samples of African affinity (Ben-Avraham et al., 1995; Hartnady et al., 1992; Mougenot et al., 1986). As stated above, the crustal nature of the MCP and NNV has been the subject of many hypothesis. Newly acquired deep seismic data within the scope of the PAMELA project attest now that thick continental crust floors the area (Moulin et al., 2020). This manuscript also discusses the nature, extent and age of a thick

volcano-sedimentary basin that overly the basement. Its roof forms a strong acoustic basement on seismic profiles which consists of basalts drilled at the bottom of most industrial wells. Given the presence of Neocomian sediments on top of these volcanics, the volcano-sedimentary basin is a pre to syn-rift formation. Following the formation of the southern Mozambique margins an intense post-rift tectonic activity also affected the area. Long term subsidence is attested by the thick cover of Cretaceous to Neogene sediments over the MCP and NNV which, however, was perturbed by several tectono-volcanic events

(Baby, 2017). Cenozoic to modern extension has also been emphasized within the MCP (Mougenot et al., 1986) and possibly southward in the NNV (Wiles et al., 2014). These deformations might be driven by the southward propagation of the western branch of East-African rift system thus paralleling the previously recognized eastern offshore branch (Mougenot et al., 1986). Both of them may connect through the northern MB (Deville et al., 2018) and delineate actual micro-plates boundaries (e.g., Saria et al., 2014).

## 3   Seismic acquisition and processing

In 2016, the PAMELA MOZ3-5 cruises onboard french R/V Pourquoi Pas? acquired a total of seven coincident multi-channel and wide-angle seismic profiles over the MCP, NNV and LM area (Figure 1; Moulin et al., 2020; Moulin and Aslanian, 2016; Moulin and Evain, 2016). This study concentrates on the 500 km long MZ3 profile trending SW-NE offshore the MCP. It runs from the NE edge of the NNV and crosses the LM towards the northwestern corner of the MB. To illuminate the deep

crustal structure thirty-two four-components Ocean Bottom Seismometers (OBS) from Ifremer's Marine Geosciences Unit were deployed every 7 nmi. 2960 shots were recorded by those instruments and by a 4.5 km, 720 channels, towed marine streamer. MZ3 crosses four out of the seven profiles acquired during the MOZ3-5 cruise: MZ2 (Verrier et al.) and MZ6 (Schnürle et al.) acquired over the NNV and, MZ4 and MZ5 that also crosses the LM (Watremez et al.). Two other profiles acquired in the NNV, MZ1 and MZ7 are presented by (Leprêtre et al., a, b).

### 3.1   Multichannel Seismic (MCS) data

Ifremer's SolidQC software was initially used for processing MCS data. It allows data quality control, 2D geometry setup and SEG-Y file generation. Further processing was performed using CGG-Veritas Geocluster software. This sequence included external mute of direct and water wave arrivals, large band filtering (1-8-64-92 Hz), predictive deconvolution (440 ms operator, 252 ms IO, 32 ms gap), 4 ms resampling, seafloor multiple attenuation using Surface-Related Multiple Elimination (SRME)

and RAMUR in the tau-p domain, time variant band-pass filtering, and pre-stack Kirchhoff migration. Velocity picking in





done after each major processing step in order to refine the final velocity model and build a coherent pre-stack Kirchhoff time migrated section (PSTM) as shown in figure 2a for MZ3 profile.

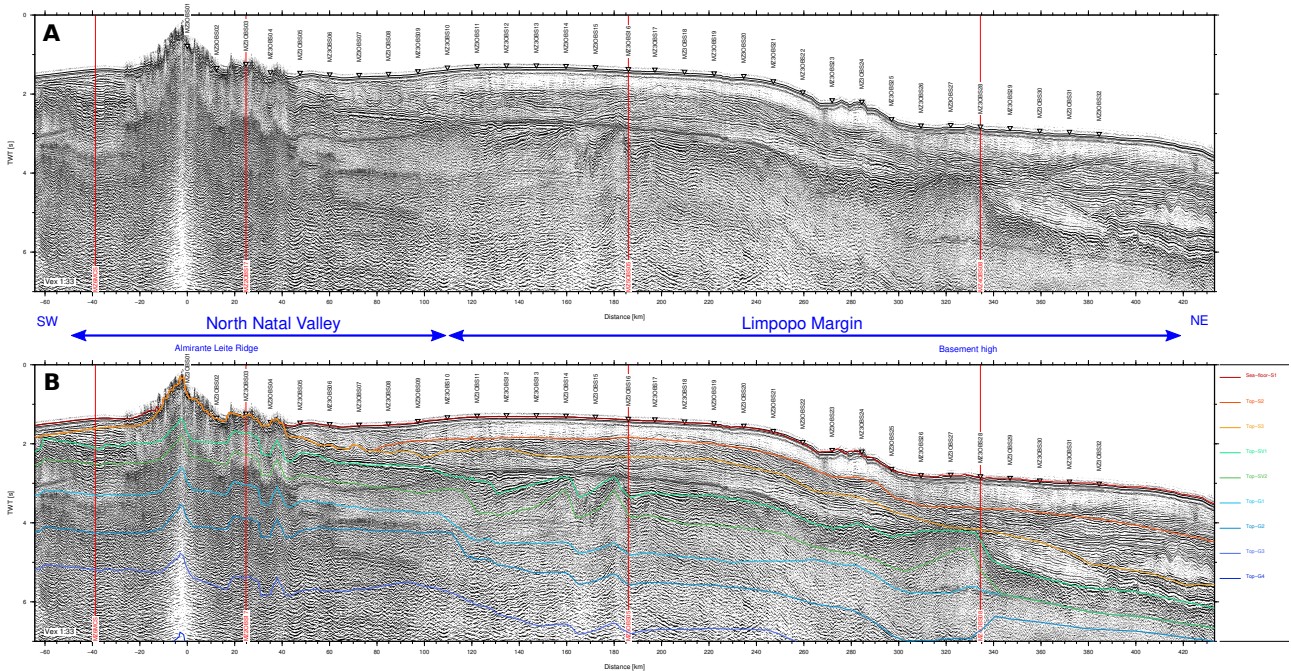

**Figure 2.** (a) Two-way travel-time record section of MCS data along MZ3 profile. (b) Same as (a) overlain by time converted interfaces from coincident wide-angle velocity model. Intersections with other profile from MOZ35 cruise are indicated by red line. OBS locations are indicated by inverted triangles.

The first hundred of km of MZ3 profile runs over the NNV and are dominated by volcanic edifices of the Almirante Leite Ridge (ALR) which completely masks the underlying structure (-20 and 40 km). On each side of the blind zone, a maximum
of 1 s.twt of upper sediments is present. These consist of a first layer (S1) of weak signal and poorly resolved horizons below the seafloor. It is very thin to the SW of the ALR but thickens on its opposite side. At the bottom of S1 a stronger reflector (top-S3) is flat and parallel to the seafloor on the SW of the ALR but is dipping and undulating towards the NE. A few strong amplitude and unconformable reflections exists on each side of the ALR beneath this interface. A set of more continuous and coherent horizons marks the top of acoustic basement (top-SV1) which appears highly and steeply fractured to the SW of the
ALR. At least 0.5 s.twt of parallel layering can be identified within the upper basement (SV1) on this side while more chaotic





and incoherent signal dominates on the NE side of the ALR. Because energy is lost below about 3 s.twt the SV2 and deeper layers are constrained by OBS data.

With increasing model distance, the seafloor over the LM forms a slight dome between 120 and 260 km and then continuously deepens. S1 continues from the NNV as a 0.5 s.twt thick layer which internal structure is almost transparent because it is

poorly resolved by our seismic. Below, S2 is a layer of intense but highly disrupted reflectivity revealing various sedimentary features, possibly channel incisions (100km model distance), contourites (360-400 km) and mass transport deposits (400-430 km). This contrasts with the S3 layer where more continuous and organized horizons are present, though of weaker amplitude. The top of the acoustic basement (top SV1) corresponds to a set of parallel and stronger amplitude reflectors in continuity with the NNV up to 300 km model distance. There a transition to a rugged surface marked by a broad basement high occurs. To

the SW of the high the basement internal structure (SV1) shows clear parallel or dipping reflectors while to its NE it is chaotic with incoherent signal. There are two trends in dipping reflectors: one to the NE slowly dipping toward the western edge of the basement high (240-260 km) and a second, dipping to the SW, that coincide with highly fractured and sharp basement. Only the internal structure of the SV1 layer, about 0.5 s.twt thick, is imaged by the MCS. No energy reflects further down and, SV2, crystalline basement (G1-G4), the Moho (top-Moho on figure 2b) are constrained from wide-angle seismic only.

**3.2  Wide-angle seismic data**

OBS data pre-processing included clock drift corrections, instruments localization using direct water wave arrivals to correct for the drift from their deployment position and eventually band-pass Butterworth filtering to improve travel time arrivals identification and picking.

Combining MCS and OBS data we identified four geological units along MZ3. From top to bottom they consist respectively

of (a) the upper sedimentary package (orange/yellow Ps phases); (b) the volcano-sedimentary sequence that forms the acoustic basement on the MCS (green Psv phases); (c) the crystalline basement (blue Pg phases); and (d) the upper mantle (magenta Pn phases). Here we describe each unit successively and how refracted and reflected phases we identified on OBS records allow to constrain them. We selected key OBS sections to illustrate this but all the records can be found in Supplementary Materials with their corresponding synthetic sections, travel times picks and fits, ray-tracing through the final model and the coincident

portion of MCS line.

On most OBS records, 2 to 3 refracted and associated reflected phases describe the sedimentary package (e.g. orange phases on figures 3A-3D). Those refracted phases have apparent velocities and offsets rarely exceeding 3 km/s and 20 km respectively. They are visible as first arrivals at very short offsets then generally extend as secondary arrivals. Corresponding reflected phases are generally well observed at short offsets inside the cone formed by direct wave arrivals (red phase). Their travel times are

coherent with those of the main horizons seen on the MCS profile (top-S2 and top-S3 on Figure 2b). Clear examples of sedimentary phases can be observed on MZ3OBS14 and MZ3OBS30 record sections (Figures 3B and 3D) located on the LM. Sedimentary phases recorded on instruments located on top or near the ALR are highly heterogeneous. The velocity structure is difficult to model and thus poorly constrained. As shown on MZ3OBS03 (Figure 3A) there are high apparent velocities at short offsets followed by a shadow area. This is modeled by a negative velocity contrast reflecting the presence of high velocity





**Figure 3.** Sedimentary and volcano-sedimentary seismic phases on A)MZ3OBS03, B)MZ3OBS14, C)MZ3OBS20, D)MZ3OBS30. For each instrument 6 panels display a) seismic record; b) seismic record with color-coded predicted arrivals; c) synthetic section with color coded predicted arrivals; d) color coded picked travel-times with uncertainty bars overlain by dotted predicted times; e) color coded seismic rays; f) MCS time migrated section with color-coded model interfaces. On a, b, c, and d, travel-time is reduced by a velocity of 5 km/s.



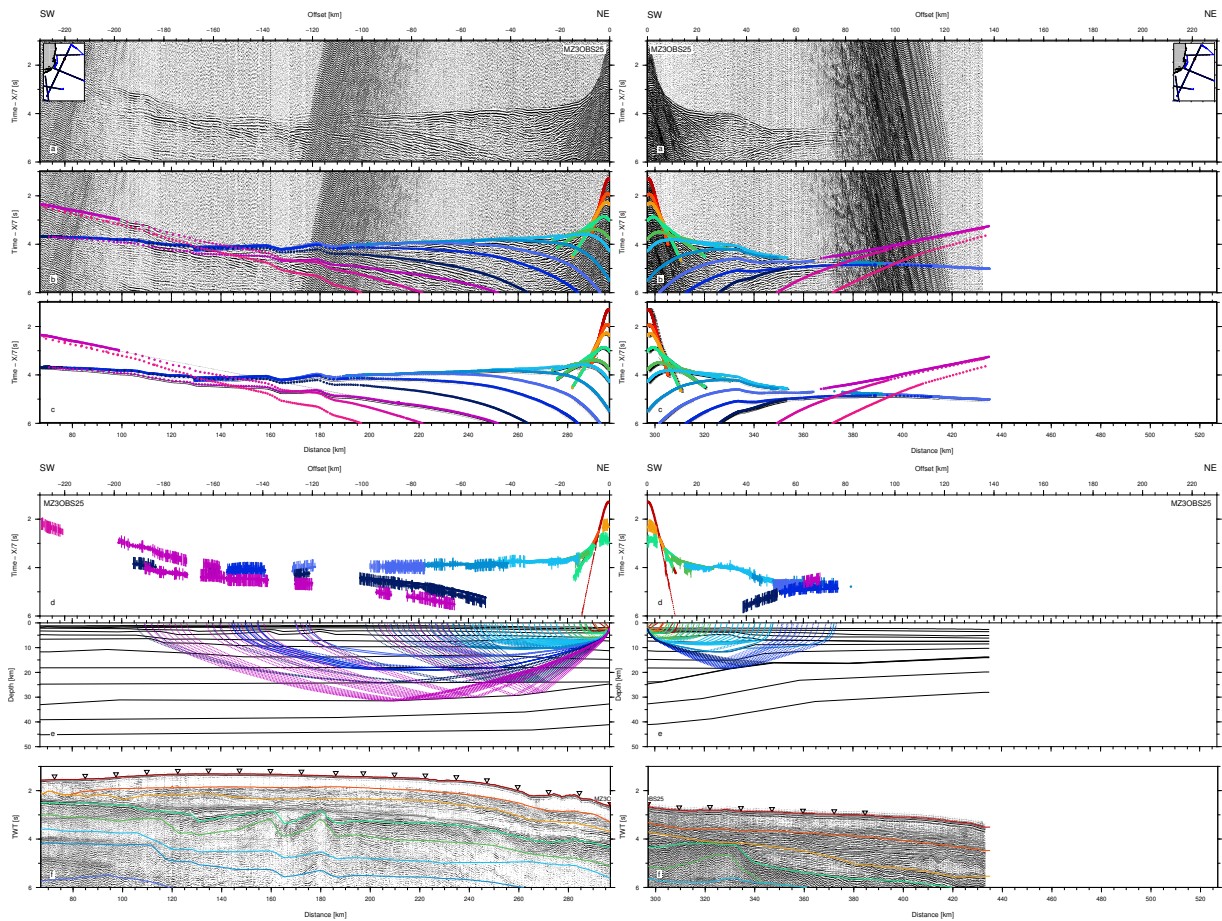

**Figure 4.** Seismic phases from the crust and mantle on MZ3OBS25. Left panels represent negative offsets (toward the SW) and right panels positive offsets (toward the NE). On each side 6 panels display a) Seismic record; b) Seismic record with color-coded predicted arrivals; c) Synthetic section with color coded predicted arrivals; d) Color coded picked travel-times with uncertainty bars overlain by dotted predicted times; e) Color coded seismic rays; f) MCS time migrated section with color-coded model interfaces. On a, b, c, and d, travel-time is reduced by a velocity of 7 km/s.

extruded volcanic material on top of lower velocity sediments. The top of the acoustic basement is marked on OBS records
by a strong amplitude reflected phase (green Top-SV1), generally the latest visible within the water cone which correlates
well with observations made on MCS (Figure 2b). The amplitude contrast is created by a strong velocity jump of at least 1.5
km/s. Instruments located over the NNV and the LM west of the basement high indicates the presence of two layers within
the acoustic basement. There is a first set of associated refracted/reflected phases (SV1) with apparent velocities lower than
5 km/s and a second set (SV2) with velocities higher than 5 km/s. (e.g. MZ3OBS20 on figure 3C). The first of these layers
when converted to two-way-time coincides with the deep layering previously described on MCS (SV1 layer on Figure 2b).
However, its base and the SV2 layer below are entirely constrained by wide-angle data. To the NE of the basement high, OBS



evidence a clear change in the characteristics of the acoustic basement internal structure as already suggested from its facies on MCS. Here a single layer (SV1) continues while SV2 is absent below the basement high. Apparent velocities are between 5 and 5.5 km/s (e.g. MZ3OBS30 on figure 3D) which are intermediate between those of SV1 and SV2 beneath the NNV and LM. Therefore, despite its prolongation toward the northeastern end of the line, the basement high appears as a clear marker of a change in nature of the acoustic basement.

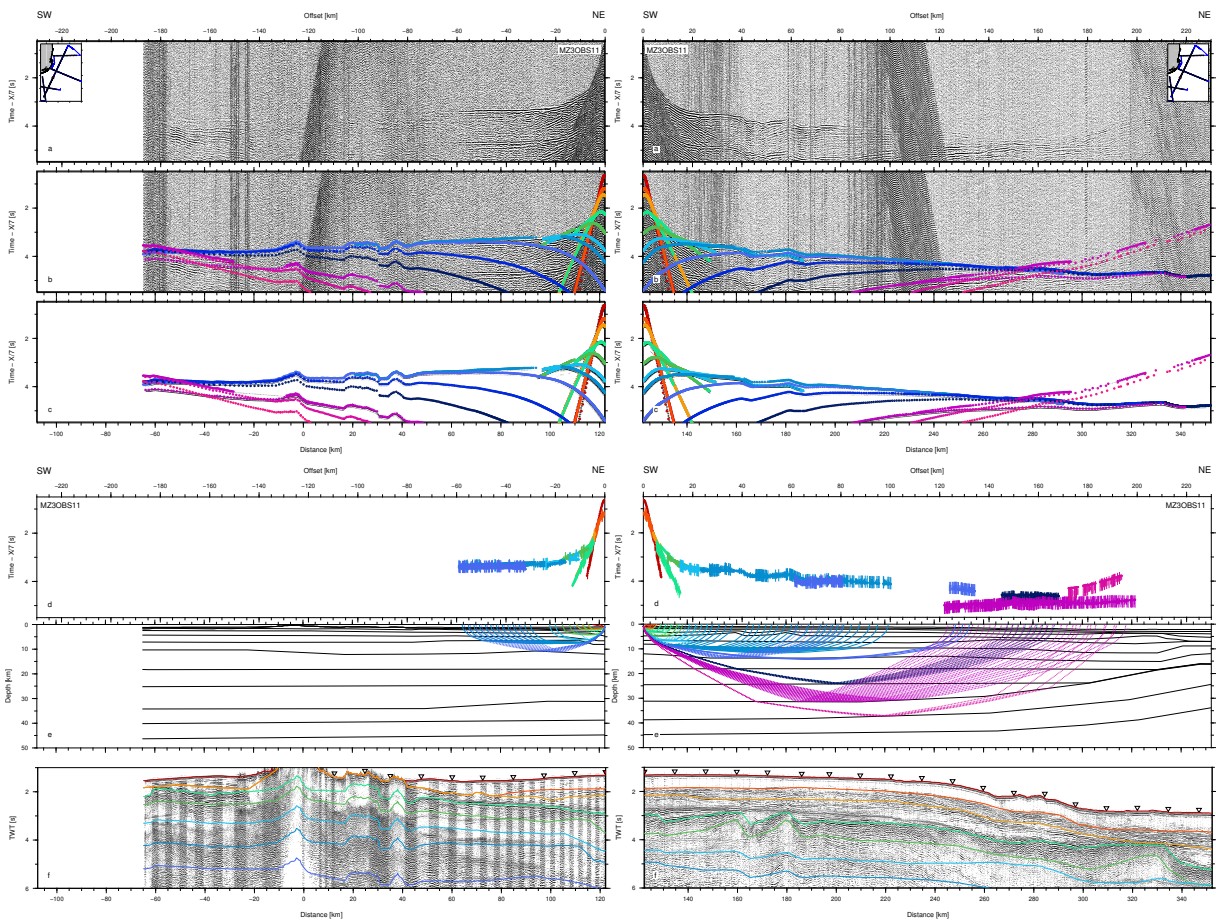

**Figure 5.** Same as figure 4 for MZ3OBS11

The crystalline basement of the LM in the central portion of MZ3 is very well constrained by OBS records. They show long offsets refracted phases with only mild variations in apparent velocities (blue Pg phases). These are associated with numerous and quite high amplitude reflected phases (blue Top-Pg phases). These combined observations suggest a thick crust made characterized by a low velocity gradient but a strong internal reflectivity. At least 5 layers can be identified based on an average account of triplications between refracted and reflected phases. Apparent velocity of refracted phases ranges from 6.5 km/s for the shallow crust to 7 km/s or slightly above for the deep crust (e.g MZ3OBS11 and MZ3OBS25 on figures 4 and 5).





Triplication between crustal and mantle phases occurs typically at about 150 km offset. This offset decreases to less than 100

km for instruments located close to the NE end of the profile (e.g MZ3OBS31 on Figure 6) implying important crustal thinning.

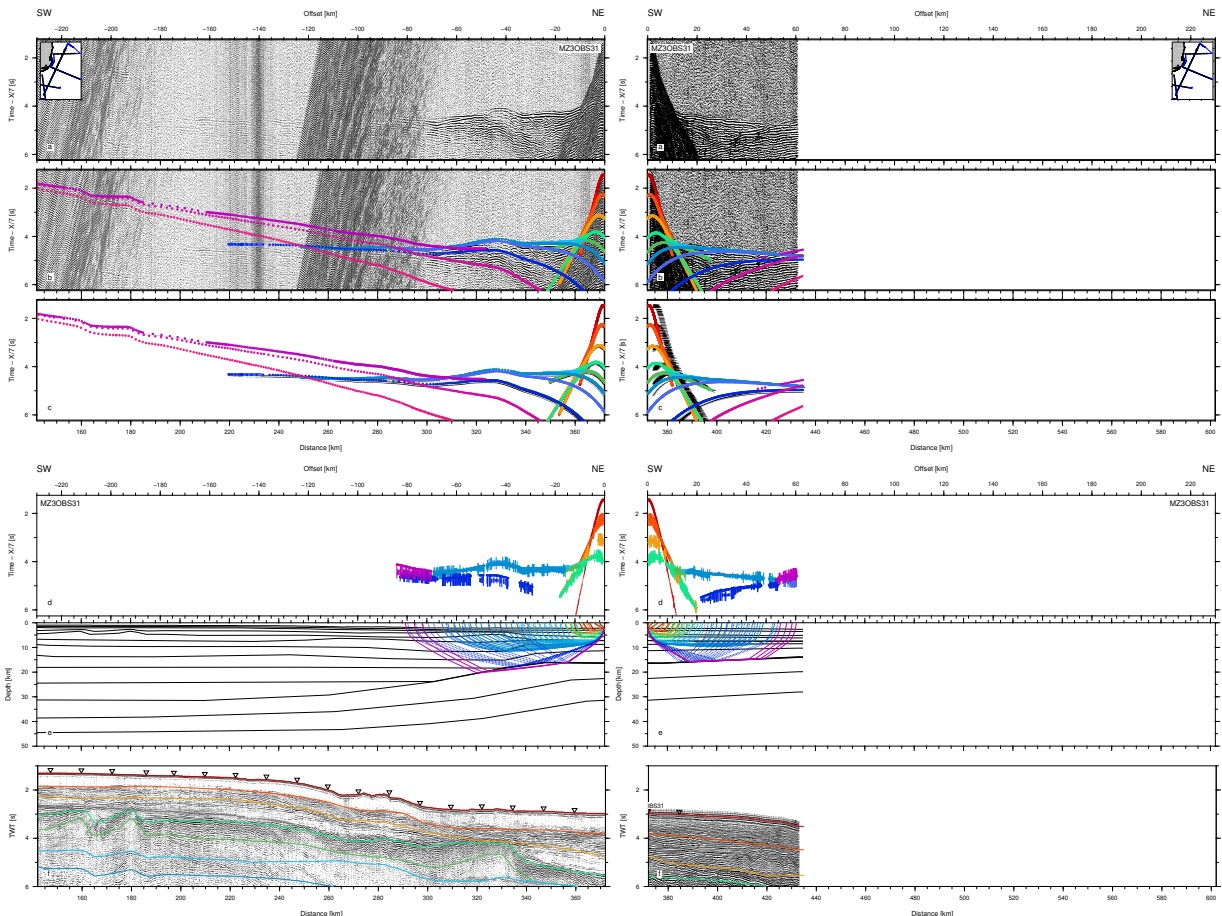

**Figure 6.** Same as figure 4 for MZ3OBS31

A different pattern of crustal phases is indeed observed on OBS records from the NE extremity of the LM. Crustal refracted phases at short offset suggest a stronger gradient and an average apparent velocity around 6.5 km/s. The gradient reduces at greater offsets but apparent velocities do not exceed 7 km/s. Some reflected phases can be distinguished but overall there is no more evidence for intense crustal layering, the most prominent corresponding to the Moho. The transition to mantle phases

occurs at a relatively short offset (about 50 km on MZ3OBS31; Figure 6) suggesting thin crust. As for the acoustic basement, all these features combined depict a clear change in crustal nature.

The deep velocity structure beneath the NNV is only partly constrained because volcanism at the ALR creates seismic blind zone. Moreover, our acquisition geometry does not allow for optimal ray coverage beneath the edifice with OBS only located on top and to the NE of the ridge. Thus the two upper crustal layers of the model are only reasonably well resolved, with

clear refracted and reflected arrivals of apparent velocities around 6 km/s and 6.5 km/s. The deeper crustal structure is poorly





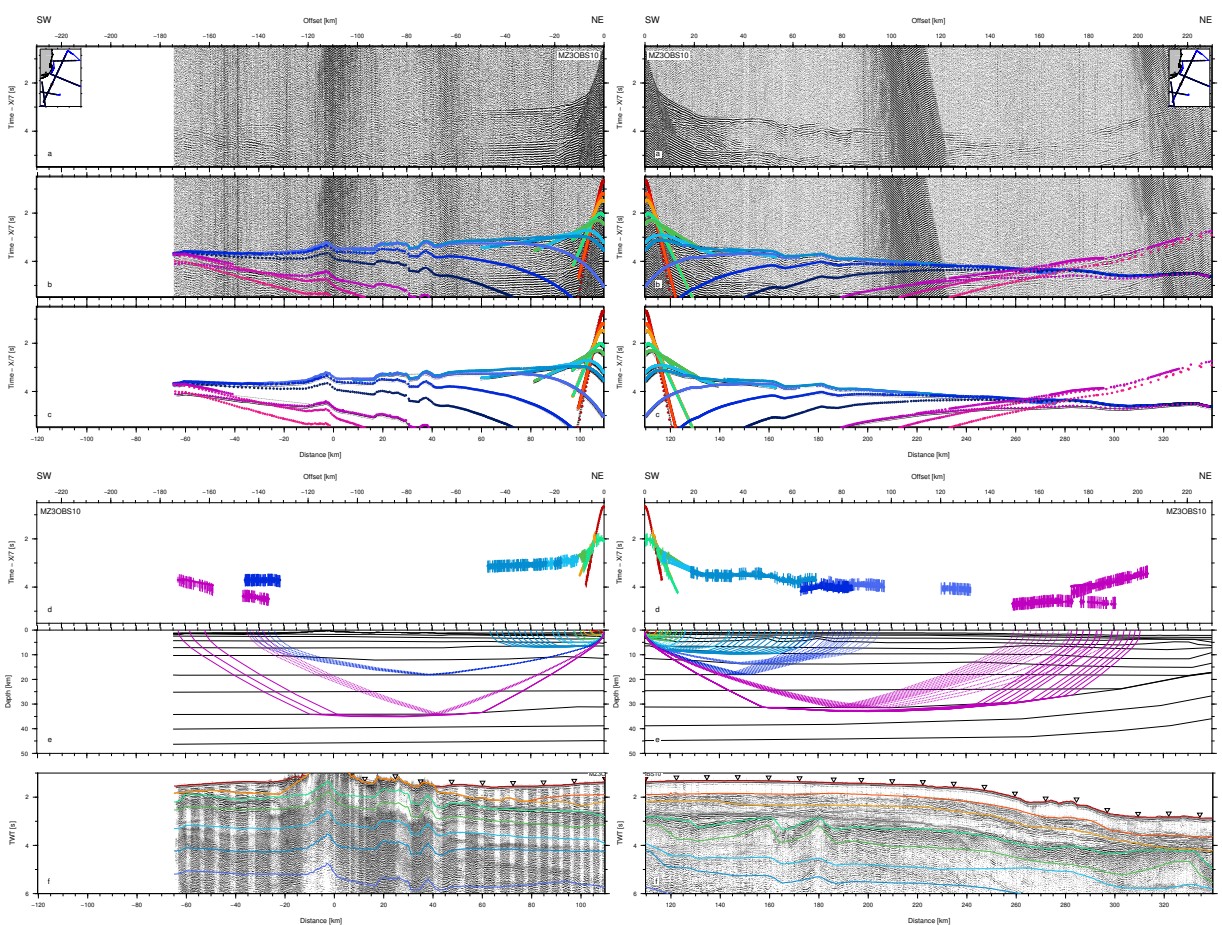

**Figure 7.** Same as figure 4 for MZ3OBS10

constrained (see MZ3OBS11 on figure 5). Though refracted phases cannot be identified, some higher amplitude phases can be distinguished at specific offsets ranges. Those can be explained as resulting from reflected waves from a layered medium similar to the one modeled for the LM. We can further interpret some far offset phases (e.g. MZ3OBS10 on figure 7) as reflection from the Moho if considering a depth of 35 km for this interface as suggested by the modeling of the crossing MZ2

and MZ6 profiles (Verrier et al.; Schnürle et al.). Nonetheless the poor constraints, we can reasonably consider the deep crustal structure beneath the NNV and ALR as similar to the western part of the LM up to the basement high. Our dataset reflect for both domains a thick crust with strong internal reflectivity, a low velocity gradient and high bottom velocities (7.3 km/s).

Arrivals from the mantle are observed on numerous OBS records throughout the profile. With many reflected phases converging towards the triplication point between crustal and mantle phases due to low velocity gradient within the crust, there

is generally high amplitude signals at this place on OBS records, making the identification of the starting point of the mantle refracted phase difficult. The consequence might be some larger uncertainties on the depth of the Moho and uppermost mantle velocities. However, these can be balanced using constraints from crossing profile MZ2 (Verrier et al.), MZ4 and MZ5 (Wa-



tremez et al.) which agree at a maximum of +/- 2 km on the depth of the Moho. Refracted mantle phases show increasing apparent velocities from 8.0 to 8.3 towards the NE (Figures 4-7). Travel time fit for these phases is not perfect and uneven from

one records to another which might also suggest important mantle heterogeneity along the profile. This heterogeneity is further evidenced by the presence of high amplitude mantle reflected phases on some records. Their dip are difficult to model with horizontal layering only. They may reveal local highly reflective body diffracting seismic waves. Anyhow, this suggest that the heterogeneous and highly reflective character of the crystalline basement might continue downward within the mantle of NNV and western LM. Again we do not have evidence for such mantle reflectivity below the thin crust in the NE extremity of the

LM but we lack of offset on these OBS records.

### 3.3    Forward modeling and travel-time inversion

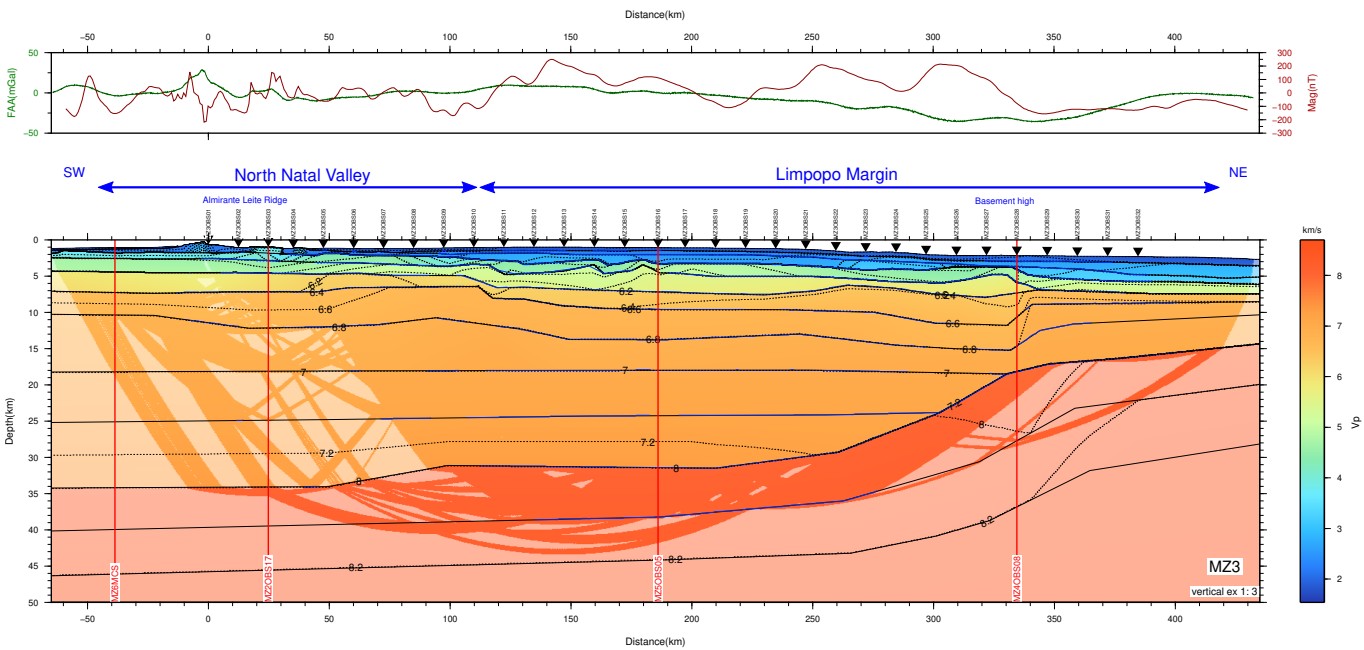

**Figure 8.** MZ3 Final P-wave velocity model with region constrained by seismic rays highlighted. Blue lines indicates where wide-angle reflections constrain the model interfaces otherwise in black lines. Inverted black triangles marks OBS positions. Vertical exaggeration is 1:3.

From MZ3 OBS data, a total of 72750 arrival times were picked from the phases described above and summarized in Table 1. Travel-time uncertainty for each phase was automatically calculated based on traces signal to noise ratio. They range from 0.025 s for high ratio to 0.25 s for poor ratio.

We used the iterative procedure of two-dimensional forward ray-tracing followed by the damped least-squares travel-time inversion of the RAYINVR software (Zelt and Smith, 1992). Our modeling proceeded following a top-to-down strategy of arrival times fitting of both wide-angle reflected and refracted phases identified on OBS data. Model interfaces were inserted when a velocity change was observed in apparent refracted velocity which mostly coincide with high amplitude reflective





events. By converting our velocity model from depth to two-way travel time we checked and adjusted the geometry of the
upper sedimentary and acoustic basement interfaces as identified on the MCS section (see Figure 2b).

MZ3 final velocity model (Figure 8) consists of 13 layers beneath the seafloor: 3 for upper sediments (S1-S3), 2 for the
volcano-sedimentary sequence (SV1-SV2), 5 for the crystalline basement (G1-G5) and 3 for the mantle (M1-M3). This model
explains 89% (64900) of the events picked with a global RMS travel-time of 65 ms which given the uncertainty assigned for
each phase results in a normalized chi-squared of 0.65. Travel-time RMS and chi-squared values for each phase and each
instrument are detailed in Table 1 and Table 2 respectively.

### 3.4 Model Evaluation

Figure 9 presents four indicators on the quality of MZ3 velocity model based on wide-angle data only. Interface depth node
spacing as well as velocity node spacing are indeed key to model the lateral variations of the seismic velocity with sufficient
resolution, but without introducing complexity not required by the data. Note that the four indicators are not calculated for the
upper sedimentary package and the top of SV1 as those layers are constrained by coincident MCS data. Their topography is
directly sample on the MCS profile while velocities are modeled from wide-angle phases.

Logically the density of velocity and depth nodes as well as the number of reflective segments are higher in the central part
of the model and decreases towards its edges and with depth (Figure 9a). Indeed these regions of the model are less sampled by
rays (Figure 9b) usually from a single direction and data quality is either poorer or degrade naturally with increasing offsets.
Because the model parameterization have been adapted to these limitations MZ3 shows overall limited smearing (+/- 3, Figure
9c) and very good resolution (above 0.9, Figure 12d). The quality of the modeling decreases at depth and towards model
extremities. Resolution values remain, however, higher than 0.5 which are still considered acceptable. Greater smearing occurs
essentially in the lower crust where only few refracted rays travel. Those have been difficult to identified on OBS records
because many phases converge around to the triplication point with the PmP and Pn (see 'Wide-angle seismic data' section
above).

### 3.5 Gravity modeling

We tested the gravimetric response of our final model against the measured (in green) and satellite-derived (in yellow and red)
free-air gravity anomaly along the profile (Figure 10). A 2-D model consisting of homogeneous density blocks was constructed
from the MZ3 final velocity model by converting P-wave velocities to densities according to (Ludwig et al., 1970) empirical
function. The resulting density ranges from 1800 kg/m3 to 2400 kg/m3 in the upper sedimentary units, 2500-2600 kg/m3 in
the volcano-sedimentary sequence, 2700 to 3000 kg/m3 in the crystalline basement and 3150 kg/m3 in the mantle. The model
was extended down to 100 km where isostatic compensation may be reached, and 500 km laterally on each sides to avoid edge
effects.

The gravimetric anomaly is relatively flat along MZ3 profile and does not exceed +/- 30 mGal (Figure 10b). There is sharp
positive peak at the ALR and a broad negative one beneath the basement high of the LM which reflect the change in crustal
nature and structure. Between these two peaks the lithosphere appears in isostatic equilibrium. The calculated gravity anomaly



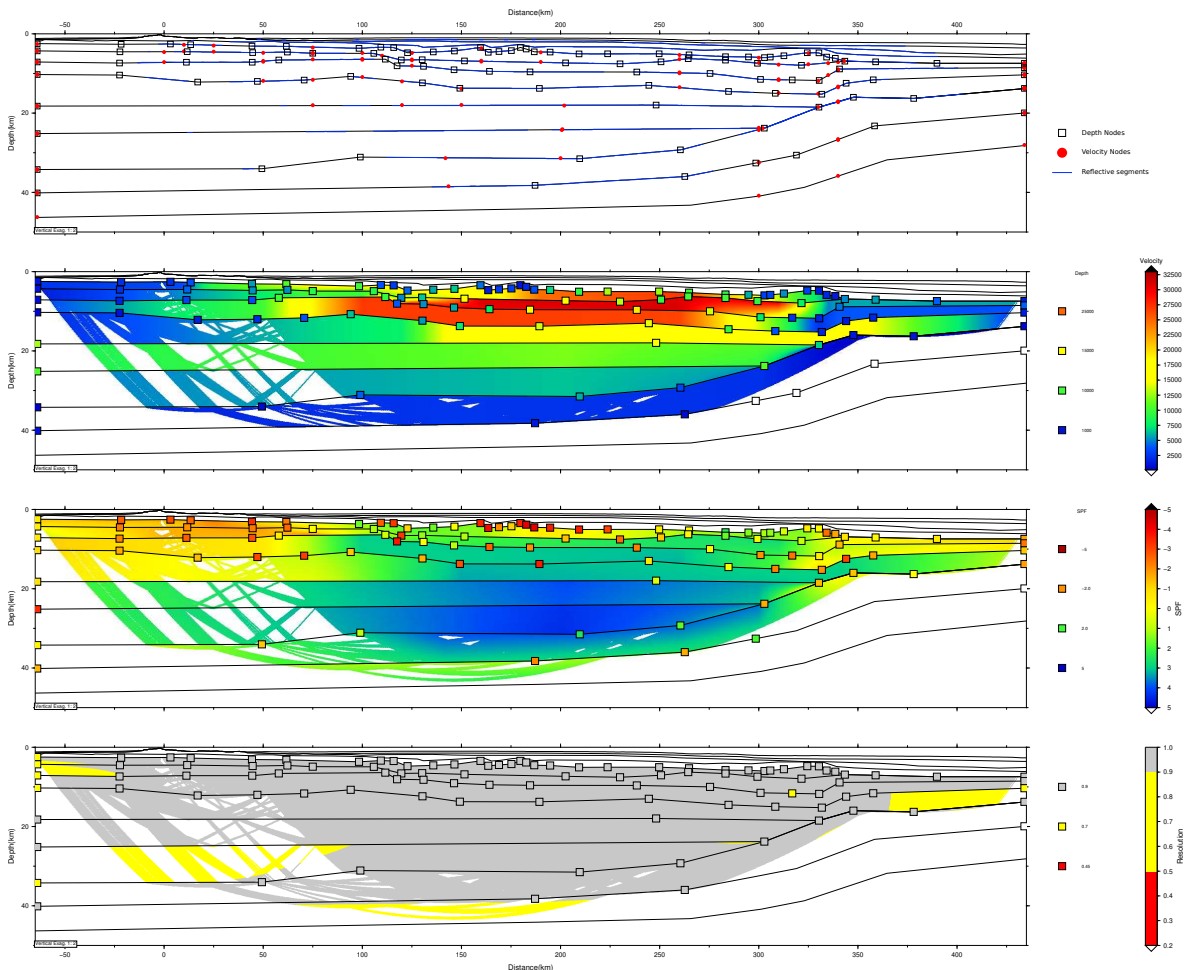

**Figure 9.** Evaluation of the wide-angle model MZ3. a) Model parameters including interface depth nodes (squares), top and bottom layer velocity nodes (red circles); interfaces where reflections have been observed on OBS data are highlighted in blue. b) Hit-count for velocity (colored) and depth nodes (squares). c) Smearing from Spread-Point Function (SPF) for velocity (colored) and depth nodes (squares). d) Resolution of velocity (colored) and depth nodes (squares). Zones that were not imaged are blanked.

based on the conversion of MZ3 velocity model reflects well the overall shape of the gravimetric anomaly all along the profile. MZ3 do not generate a broad negative anomaly at the transition beneath the basement high of the LM but the change in crustal properties generates a clear jump from negative to positive values consistent we the observed gravity anomaly. Note that we had

to lower the densities within the mantle layers from 3300 to 3150 kg/m3 in order to adjust the measured gravimetric anomaly which were showing the correct trend but with lower values to the SW and higher values to the NE.







**Figure 10.** Gravity modeling along MZ3 profile. a) Density model up to a depth of 50 km overlain by interfaces from wide-angle modeling. b) Free-air gravity anomaly derived from satellite altimetry (Pavlis et al., 2012) along profile (red) and at 10, 20 and 30 km laterally (yellow lines); measured during the MOZ35 experiment (blue line) and calculated from MZ3 velocity model (green line).

### 3.6 Uncertainties estimation

Velocity and depth uncertainties of MZ3 final velocity model were estimated using the Vmontecarlo code (Loureiro et al., 2016). Vmontecarlo explores the model space by generating random models. It evaluates the ability of each model to fit the observed data set and translate it to estimates of uncertainties given some quality thresholds.

For computational cost, the explored model space was reduced by limiting the number of parameters and fixing some bounds. For MZ3 we chose to maintain fixed sedimentary and volcano-sedimentary layers as well as basement depth nodes because those benefit from refined MCS constraints. This results in a total number of variable parameters of 148 as shown in figure B1.





**Figure 11.** Global uncertainty plots for MZ3. a) Maximum and b) minimum admissible velocity deviations from the preferred model, built from 126 models within the thresholds defined in the text. Shaded areas indicate ray coverage. Preferred model's interfaces are indicated by black lines and velocity deviations are colored according to color scales.

We also allowed a maximum fluctuation of +/-0.50 km/s on velocity nodes while the maximum depth node variations were
set to bands of 1, 2 and 3 km for the upper crustal, the 3 mid-crustal and the Moho interfaces respectively. We further limited
the search by generating a maximum of 50000 random models and imposed a scaling factor for velocity and depth bounds
that, starting at 20% of their maximum value, progressively increased to 100% within the first half of the randomly generated
models. Finally the total number of observed travel times was decimated to be less than 50000 picks to reduced ray tracing
computing time.

Scores are calculated for each model according to a function that takes into account travel time fit and the ratio of traced rays
to the number of observed events (Loureiro et al., 2016). MZ3 final velocity model gives a score of 0.88 with 43928 rays traced





over the 49725 picks, an RMS of 0.07 s and, after adjusting the pick's uncertainties, a chi$^2$ of 1. Among the 50000 randomly generated models within the bounds given above, 46110 were valid and scored up to 0.81. Supplementary Materials give a summary of this model space by plotting every 20 km model distance all 1D velocity-depth profiles color coded according to

their normalized average score. Settings a confidence threshold at 95% of the maximum normalized average scores local depth uncertainties for a particular velocity can be estimated and vice versa. Vmontecarlo analysis shows that velocity uncertainties are globally stable around 0.1 km/s both along the model and with depth. Similarly, on average depth-uncertainties are quite stable along the profile. They increase with velocity from 0.5 km at 6.5 km/s to 1.5-2 km at 8 km/s. Higher uncertainties values are observed around 300 km model distance because of the sharp transition in crustal structure and the potential lateral

effects due to the orientation of MZ3. It is interesting to note that depth uncertainties for a velocity of 7 km/s is usually high over a couple of km to a maximum of 5 km in a few places. Up to 300 km model distance this can be explained by the very low velocity gradient required in mid/lower crust. In fact Vmontecarlo analysis further confirms that most of the basement is composed of material with velocities higher than 6.5 km/s even towards the NE end of the profile (distance greater than 320 km).

In order to visualize uncertainty estimates on profile a subset of possible alternative models were selected. These models respect the following criteria we judge acceptable: a score over 75% of the preferred model's score, a chi$^2$ lower than 2, a RMS lower than 0.1 s and at least 80% of the rays traced by the preferred model. This subset represents 126 models that were combined to produce the minimum and maximum admissible velocity deviations maps shown in figure 11. Large uncertainties within the depth bounds allowed for interfaces are expected when large velocity contrast exist and should be ignored. Globally

outside these hatched areas velocities can vary from +/- 0.10 km/s to +/- 0.15 km/s with rare excursions over +/-0.2 km/s.

### 3.7 MCS data pre-stack depth migration (PSDM)

A pre-stack depth migration and a residual moveout analysis were performed in order to convert MCS data from time to depth and verify the accuracy of the wide-angle velocity model. The Kirchhoff pre-stack depth migration procedure is described in details by Schnürle et al. and the resulting section for MZ3 profile is shown in figure C1a. As for the pre-stack time migration

(figure 2) the full sedimentary cover is well imaged up to the top of the SV1 layer. Below, the section is mostly transparent with only intermittent and tiny events within the SV1 layer. In the common image gathers (CIGs: figure C1b), the top of Sv1 coincides with a strong low wavelength reflector, followed by poorly coherent and under migrated events. The semblance panels (figure C1c) is similarly well focused in the upper sediments and top of SV1 interfaces while it is blurred at greater depth and beneath the ALR.





## 4  Crustal structure of the Limpopo Margin

### 4.1  Seismic stratigraphy of upper sediments

Our final velocity model (Figure 8) shows two upper sedimentary layers over the NNV increasing to three layers over the LM. They are together less than a kilometer thick around the ALR and progressively thicken as the seafloor and acoustic basement deepens toward the NE. This package shows globally low velocities between 2 and 3 km/s top to bottom, reaching values slightly above 3 km/s at the northeastern end of the line where it is the thickest (3.5 km). It is therefore largely made of uncompacted terrigenous sediments except close to the ALR where strong velocity contrasts and inversions suggest interleaved volcanic layers. Over the LM, sedimentary horizons are strongly deformed showing typical features of a margin shelf and slope with incised valley or reworked sediments such as mass transport deposits and contourites (Figures 2, 12).

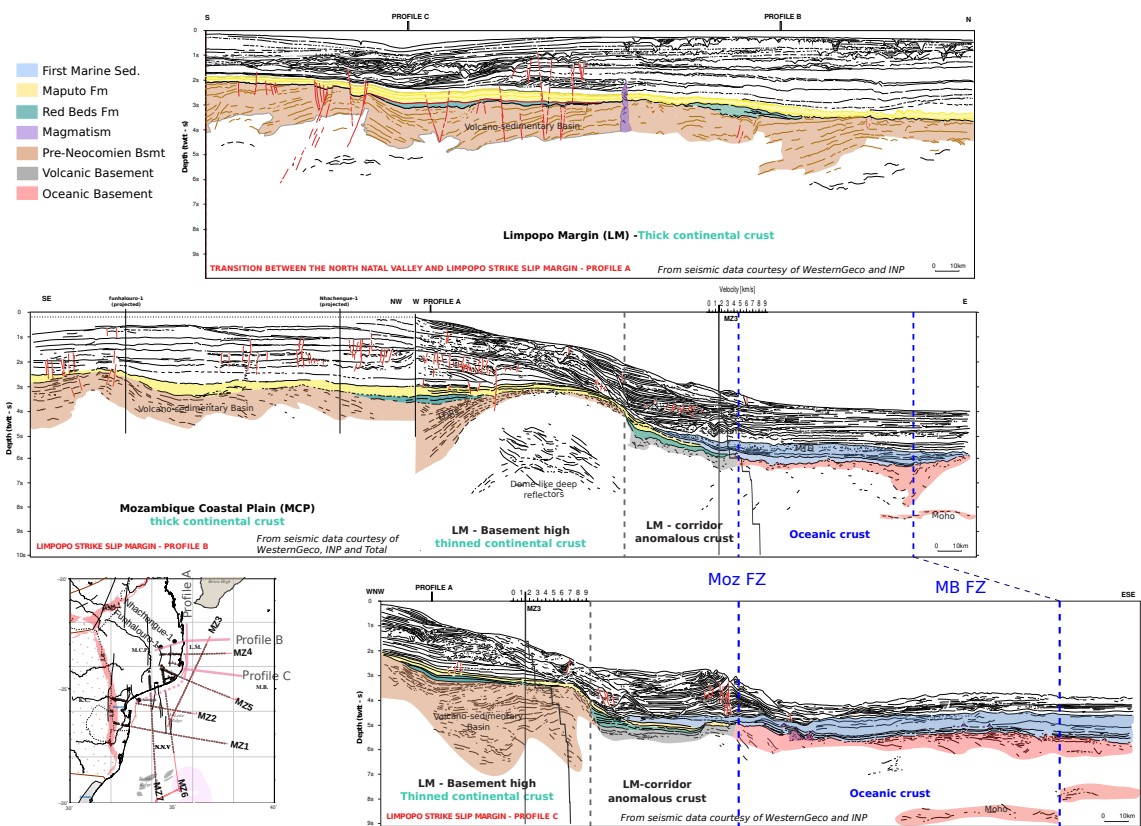

**Figure 12.** Compilation of line drawing of commercial MCS two-way travel-time profiles (see map inset for location) superimposed by stratigraphic interpretation.

On the MCP and offshore Mozambique, several wells were drilled into this upper sequence and up to the acoustic basement for some of them. Combined with commercial seismic profiles (Figure 12, 13) we performed a seismo-stratigraphy analysis





to constrain the ages of the oldest deposits which span the syn-rift and early post-rift period that are of interest for this study. More precisely we extended to the LM previous seismo-stratigraphy studies produced for the NNV (Baby et al., 2018; Schnürle et al.; Verrier et al.). Profile A is the direct north-south prolongation along the LM of a profile presented by these authors. It connects to the Sunray 1 well (Salman and Abdula, 1995) located in the NW corner of the NNV (Figure 13). We further

analyzed two profiles (B and C) than are normal to profile A and strike roughly E-W across the LM. Among them Profile B has an onshore portion that connects to Funhalouro-1 and Nhachengue-1 wells and Line SM-59 presented by Salman and Abdula (1995). Along all these lines, the horizons of late-Jurassic to Neocomian formations, namely Red Beds Fm and Maputo Fm (in green and yellow respectively on figure 12), can be confidently extended towards the LM from both the NNV (Sunray-1 well) and the MCP (Funhalouro-1 and Nhachengue-1 wells). As mentioned by Salman and Abdula (1995), these two formation

unconformably cover the acoustic basement. Red Beds are only observed filling local fossil depressions on the MCP and northern NNV (Figure 12, green deposits). A possible similar deposit is interpreted at the eastern foot of the basement high on Profiles B and C. On top of these restricted deposits or directly on the acoustic basement, the Maputo Fm appears widespread over both the NNV and the MCP. Most importantly, a continuous sequence of Maputo Fm (in yellow), akin strongly eroded, can be drawn on Profile B across the entire LM. It indeed passes clearly over to the east of the basement high where the first

marine sediments (in blue on figure 12) are seen onlapping the formation.

## 4.2  Nature and deformation of the acoustic basement

At the base of upper sediments, the top of the acoustic basement as identified on MZ3 MCS line consists of a set of high amplitude parallel reflectors over the NNV and western LM changing to a rough surface to the NE end of the line. On the velocity model (Figure 8), it is characterized by a large velocity jump (1-1.5 km/s) and shows values above 4.5-5 km/s just

below. It is relatively flat over the NNV at about 2 km depth and deepens to 3.5-4 km in the LM to finally reach 6 km passed 330 km model distance. Its highly perturbed topography evidences important deformations most probably controlled by highly dipping and near vertical faults. On each side of the ALR they bound a few horsts and grabens while over the LM they are highlighted by fan-like dipping reflectors and steep basement highs. The most prominent basement high, located at 310-340 km model distance, is clearly an isolated basement structure controlled on each side by steep faults. To its SW, the fault zone

appears wider at the basement surface suggesting a deep fault rising as a flower system (Figure 2).

Internally the acoustic basement show chaotic and incoherent signal in some places (60-120 km, 300-420 km, Figure 2) but also clear basement-parallel or dipping reflectors of variable amplitude in others (-60 to -20 km; 120-260 km). This already suggests that the top of the acoustic basement is not the roof of the crystalline basement, at least up to the basement high at 330 km. Its internal structure is indeed only partially resolved by the MCS. OBS data further constrain a 3 to 4 km thick

unit made of two layers (SV1 and SV2 layers in green on figure 2b). They are respectively 1-1.5 km and 1-2.5 km thick with velocities ranges of 4-5 km/s and 5-6 km/s on average though slightly lower values are observed beneath the ALR (Figure 8). Both seismic facies and velocities argue in favor of a deep volcano-sedimentary unit that is continuous from the NNV towards the LM but ends at the basement high at 330 km. Beyond 340 km model distance only a single layer (a prolongation of SV1





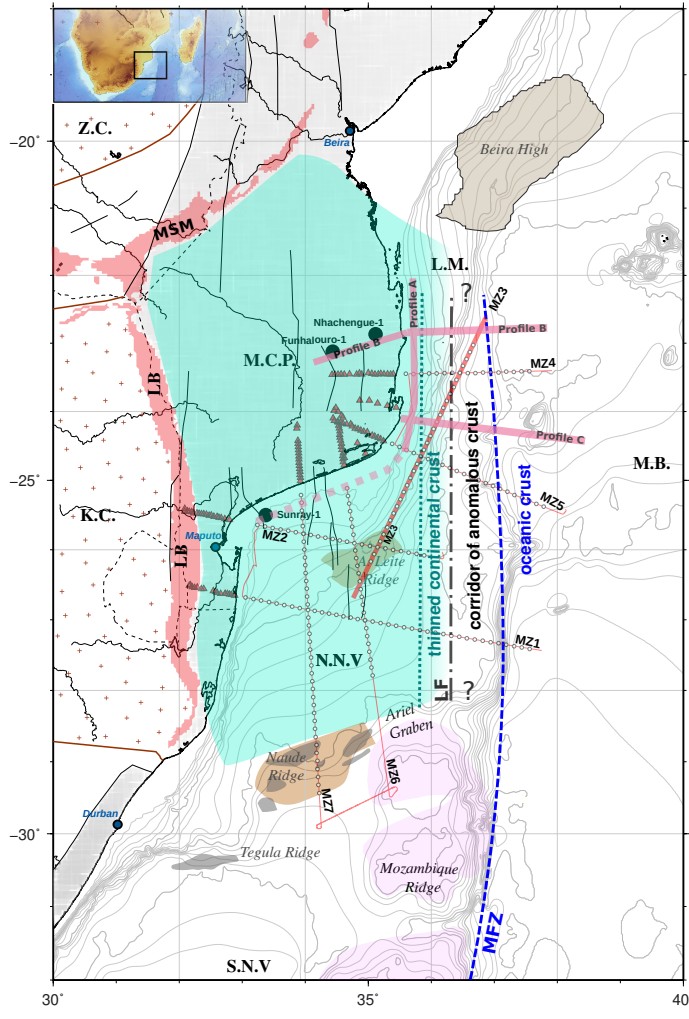

**Figure 13.** Map of the main segmentation and boundaries within our study area resulting from the combine interpretation of PAMELA-MOZ35 deep seismic profiles. Pink lines are locations of line drawings of commercial MCS profiles shown in figue 12. Background shows main onland geological units and structures: KC: Kaapval Craton; LB: Lebombo monocline; MSM: Mateke Sabi Monocline; MCP: Mozambique Coastal Plain; NNV: North Natal Valley; ZC: Zimbabwe Craton. Offshore bathymetric contours in the LM: Limpopo Margin; Mozambique Basin (MB) and South Natal Valley (SNV) and main geolocical features: Almirante Leite Ridge (ALR), Beira continental block or high (BH), Limpopo Fault (LF); Mozambique Ridge (MR), Mozambique Fracture Zone (MFZ), Tegula and Naude ridges, Ariel Graben (AG). MOZ35 seismic acquisition in the NNV and LM is shown in red with MZ3 profile highlighted in a bold red line.

for modeling purpose) remains which is 1.5 km thick and has intermediate velocities of 5.0-5.5 km/s. These velocities together
with its homogeneous, high amplitude but chaotic facies suggests instead the presence of extruded volcanism only.



Commercial MCS data and companion PAMELA profiles complement and clarify the nature and extent of the deep volcano-sedimentary unit (SV1 and SV2). As for MZ3, all those profiles depict a prominent basement high at the oceanward limit of this unit. On crossing profiles MZ5 (Watremez et al.) and profile C (Figure 12) its seismic signature clearly consists of deformed but parallel reflectors gently dipping or undulating from beneath the basement high until the western ends of these lines. On
MZ1 profile to the south (Leprêtre et al., a), these parallel and undulating horizons are seen striking through the entire NNV before they dip oceanward together with the top of the basement high and their signal is lost (i.e. below MZ1OBS11). To the north, on MZ4 (Watremez et al.) and Profile B (Figure 12) the unit clearly depicts instead a set of landward dipping reflectors (LDR) flanking the edge of the basement high. These LDR, however, connect with deep, wavy, and parallel horizons seen extending from the MCP in the west of Profile B and similarly extending all along the north-south oriented Profile A that is
contiguous with the northern NNV (Figure 12). All together this reveals that the volcano-sedimentary unit is ubiquitous to our study area and form a widespread basin that terminates oceanward as a prominent basement high.

### 4.3 Crustal nature and segmentation

Below the volcano-sedimentary basin, the top of the crystalline crust is higher at 4-5 km depth in the NNV than in the LM where it reaches 6-7 km depth on MZ3 (Figure 8). Considering the presence of a volcanic layer instead of a basin oceanward of
the basement high, the top of the crystalline crust is there deeper at 7-7.5 km depth. Along the entire MZ3 profile this interface marks a velocity jump to values above 6 km/s that is responsible for a strong reflective phase observed on OBS records. Most importantly it is the place from which the velocity gradient of the crust becomes much lower around 0.044 km/s/km compared to the 0.25-0.3 km/s/km gradient within the upper acoustic basement.

Over the NNV the five layers that make the crystalline crust represent a 30 km thick unit with velocities between 6.0 km/s
and 7.3 km/s top to bottom and a Moho lying at 34 km depth. Over the unthinned crust of the LM (up to 200 km) the same five layers represent a 24 km thick crust with a velocity range of 6.2-7.3 km/s and a 31 km deep Moho. The strong reflectivity of crustal phases observed on OBS records over both areas is responsible for the numerous layers modeled within the crust and mantle and attest of their important magmatic content. Therefore, despite slight thinning, the crustal structure of the NNV and western LM are similar and consists of: an upper volcano-sedimentary basin, a low velocity gradient within the crystalline
basement, strong magmatic intrusions, and anomalously high velocities at the base of the crust. These characteristics are also those observed and highlighted on velocity models produced along PAMELA profiles that images the MCP to the west (i.e MZ4 and MZ5: Watremez et al.)) and the NNV further south and west of MZ3: i.e. MZ1 (Leprêtre et al., a), MZ2 (Verrier et al.), MZ6 (Schnürle et al.) and MZ7 (Leprêtre et al., b). A comparison between 1-D velocity-depth (vz) profiles extracted every 10 km along MZ3 with vz profiles from MZ7 (Figure 14a-b) shows the coherence in crustal structure between the MCP,
the NNV and the LM. Compared to a world-wide compilation of continental crust of different tectonic settings (Christensen and Mooney, 1995) our vz profiles clearly shows a typical continental trend (Leprêtre et al., b; Moulin et al., 2020). The observed shift toward high velocities can be explained by a high magmatic content at every crustal levels as thoroughly discussed in these studies.




On MZ3 velocity model (Figure 8), crustal thinning occurs over a distance of 100 km to the SW of the basement high
where the Moho rises from 31 km to 16 km deep. It coincides with a slight increase in velocity at the top of the crystalline
basement (from 6.2 to 6.4 km/s) and a progressive decrease at its base (from 7.3 to 7.0 km/s). The velocity gradient remains
thus stable and low (Figure 14b) while its internal highly reflective character is still observed. Such characteristics are coherent
with those described on velocity models produced along crossing MZ4 and MZ5 profiles (Watremez et al.) and further south
along MZ1 profiles (Leprêtre et al., a). All together they emphasize an eastward prolongation and thinning of magmatically
intruded continental crust beneath the LM (Figure 13).

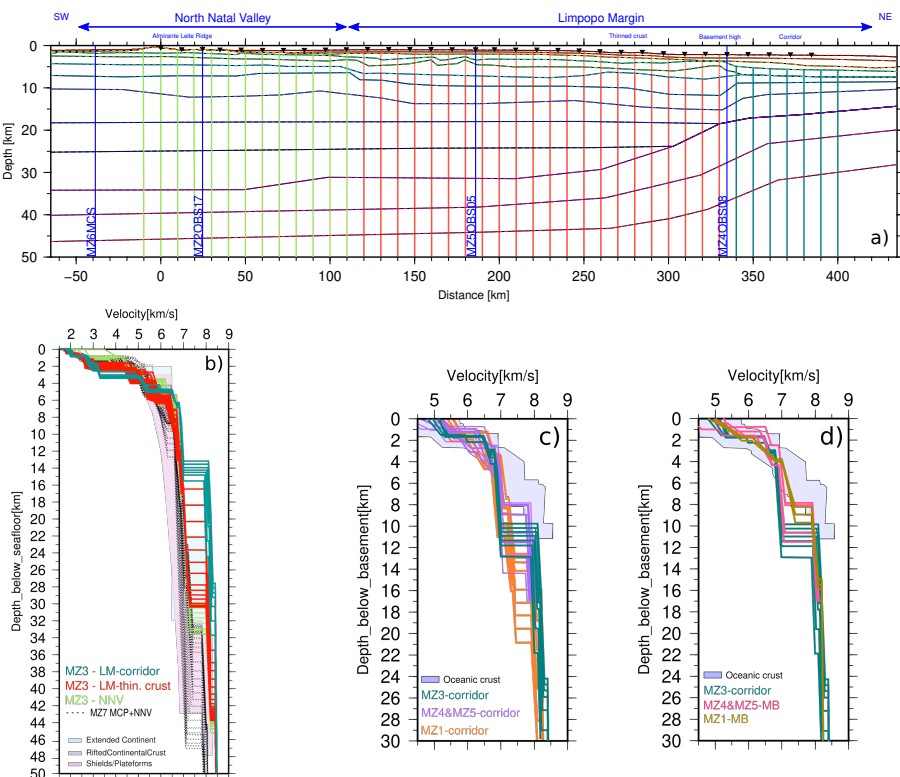

**Figure 14.** a) 1-D velocity-depth (vz) profiles extracted along MZ3 final velocity model and color coded according to main domains and
segments. b) MZ3 vz profiles below seafloor compared to MZ7 vz profiles of the MCP and NNV, and compiled vz profiles of continental
crust for different settings (Christensen and Mooney, 1995). c) Comparison of vz profiles below the acoustic basement of the corridor of
anomalous crust extracted from MZ1 (Leprêtre et al., a), MZ3 (this study), MZ4 and MZ5 (Watremez et al.) models with a compilation of
vz from 'normal' Atlantic oceanic crust (White et al., 1992). d) Comparison of MZ3 vz profiles below the acoustic basement of the corridor
of anomalous crust with vz for the oceanic crust of the MB extracted from MZ1 (Leprêtre et al., a), MZ4 and MZ5 (Watremez et al.) and
compared to the compilation of (White et al., 1992).





On the northeastern side of the basement high, the crystalline crust of the LM is only 10 km thick on MZ3 velocity model (Figure 8). It is made of 3 layers with velocities between 6.5 km/s to 7.0 km/s top to bottom that show little internal reflectivity. While the two deepest layers present a low gradient (0.025 km/s/km) similar to the adjacent continental crystalline basement, the upper layer has an intermediate velocity gradient (0.1 km/s/km) between those of the acoustic basement above and the deep crystalline basement below (Figure 14b). This anomalous trend is also observed on the 'transitional crust' identified on MZ4 and MZ5 (Watremez et al.) and east of the basement high on MZ1 velocity model (Leprêtre et al., a). Taken all together they reveal a 60-80 km wide, N-S trending corridor of anomalous crust isolated between the thinned MCP and NNV continental crust to the west and the MB oceanic crust to the east (Figure 13).

The basement high locates therefore an important segmentation in the crustal structure of the LM (Figure 13). Not only it coincides with the eastward termination of the volcano-sedimentary basin and a change in nature of the acoustic basement but it also marks a profound modification of the crystalline crust. We can further notice that uppermost mantle velocities appears normal and stable at 8 km/s on each side of the basement high while a slight velocity decrease to 7.9 km/s (Figure 8), more pronounced on MZ4 and MZ5 velocity models (Watremez et al.) is present beneath the feature. This suggest that the basement high is the surface expression of a vertical frontier that is deeply rooted in the lithosphere. As such it is comparable to the MFZ that bounds the opposite side of the corridor and separates it from the oceanic crust of the MB. We will therefore refer to this zone of strongly localized deformation as the Limpopo Fault (LF, Figure 13).

To summarize, the LM is a N-S elongated margin cut by two major fault zones that segment its crustal structure in a western continental part, a central corridor and an eastern oceanic part. The continental domain is the direct eastward prolongation of the continental crust that floor the MCP and NNV. Defining the nature of the crust within the corridor remains a challenge. Vz profiles (Figure 14c-d) reveal a mixed composition. On one hand a 3-4 km thick upper part with a typical signature of oceanic crust caused by the presence of extruded volcanics. On the other hand a lower part that still preserve the low gradient characteristics of continental crust. Its crustal thickness is also extremely variable (5 to 15 km) and appears thicker to the south (MZ1) than to the north (MZ3, MZ4, MZ5: Figure 14c). Vz of the MB oceanic crust located just east of the MFZ also show some changes with a velocity signature close to the corridor to the north where it is the oldest and a signature that shift toward typical oceanic crust to the south (Figure 14d).

## 5 Discussion

### 5.1 Geodynamic implications

Deep crustal velocity models produced from the PAMELA-MOZ35 seismic experiment have revealed the presence of a 40 km and 30 km thick crust respectively beneath the southern MCP and offshore NNV (Leprêtre et al., a; Moulin et al., 2020; Schnürle et al.; Verrier et al.). They also show a continuous and mildly deformed volcano-sedimentary cover over these two entities. These findings undoubtedly confirm the continental nature of the MCP/NNV as suggested previously by seismological and magnetic studies (Domingues et al., 2016; Hanyu et al., 2017). This work further precises the crustal architecture of the area, delimiting the eastern edge of this continental unit along the LM. The presence of an homogeneous basin over a thick





continental crust attest of an intra-continental depositional process (Aslanian et al.; Moulin et al., 2020) rather than a rifted
margin process (e.g. Cox, 1992; Klausen, 2009; Watkeys, 2002). Our tectono-stratigraphic analysis only place an upper bound
to the formation of the basin in pre-Neocomian. So far it has not been dated more precisely despite that its upper volcanic
layer was reached by several wells (see section 4.1 and Schnürle et al.; Verrier et al.). We can thus speculate that it formed
during the Karoo phase which affected the entire African continent in late Palaeozoic/early Triassic ((Daly et al., 1991).
However, this would leave a large, 30 My, sedimentary gaps between extrusions of Karoo volcanics, which might compose
the roof of the basin, and deposition of late-Jurassic to Neocomien Red Beds and Cretaceous Maputo formations. Therefore,
another hypothesis is that it formed in late-Jurassic either before or contemporaneously with the formation of the adjacent MB
(Aslanian et al.). The high magmatic content of the basin as well as evidence of strongly intruded continental crust suggest that
magmatism may have overloaded the crust and created the necessary vertical subsidence (Aslanian et al.; Moulin et al., 2020;
Tozer et al., 2017).

Our study also emphasizes the eastward prolongation and termination of the volcano-sedimentary basin along the LM where
it is strongly deformed. At depth, it is also the place where important crustal thinning and segmentation is evidenced (Figure
13). This clearly indicates that the LM is the place where rifting localized to accommodate the opening of the MB rather than
over then entire MCP/NNV. In previous scenarios based on 'tight' fit kinematic framework in which Antarctica partly overlaps
Africa, rifting was either postulated to concentrate along the Lebombo monocline or beneath the MCP/NNV depending whether
the area was interpreted as oceanic or a volcanic rifted margin (Cox, 1992; Klausen, 2009; Leinweber and Jokat, 2012; Mueller
and Jokat, 2019; Watkeys, 2002). Neither of these hypothesis are supported by our new observations.

Overall, this means that the MCP/NNV must be excluded from the Africa-Antarctica corridor (AAC in Leinweber and Jokat,
2012; Mueller and Jokat, 2019) and a 'looser' plate fit must be adopted in East-Gondwana kinematic reconstruction (Moulin
et al., 2020; Thompson et al., 2019). Such framework excludes any initial rifting phase with normal ( E-W) or oblique (NW-SE)
plates movement (e.g. Cox, 1992; Reeves et al., 2016) which, to the our opinion, has never been clearly evidenced or described.
Invoking indeed the orientation of magmatic dyke swarms to attest stress field direction (Mueller and Jokat, 2019; Reeves et al.,
2016) is highly speculative as they may be strongly controlled by inherited lithospheric discontinuities (Jourdan et al., 2006).
Similarly, the presence of a wide crustal necking zone cannot solely justify normal or oblique rifting (e.g. Vormann et al.,
2020). Indeed the LM itself shows such characteristics but given our alternative geodynamic framework it was affected instead
by strike-slip or slightly trans-tensional rifting following a continuous N-S direction of plate motions during the opening of
the MB.

## 5.2 Strike-slip rifting along the Limpopo Margin

Along the LM strike-slip rifting is emphasized by the Limpopo Fault. It forms a zone of deeply rooted and strongly localized
deformation rising to the surface as a flower structure (Figure 2,12. Basement uplift is evidenced by the presence of a prominent
high along the fault with deposition of pre-Neocomian Red Beds on either side and signs of erosion of the Maputo Sands
formation that covers it. Taking all together these indicators reflect strike-slip strain along the LF which lasted up to early
Cretaceous. Despite a thick sedimentary cover and strong magmatic content which affect the definition of seismic profiles we





identified a long wavelength wavy deformation within the volcano-sedimentary basement that looks like folding as well as a few horsts and grabens (Figure 2,12. Those are located at the western edge of the LM and might reflect some partitioning in the
deformation of the upper crust with a slight normal component. Uplifts, en-echelon pull-apart structures and shear foldings are typical features observed along highly oblique or strike-slip margins (e.g. Mascle and Blarez, 1987) but more generally those contexts are favorable to strain partitioning and a wide range of deformation features (Brune, 2014; Teyssier et al., 1995)(Brune, 2014; Teyssier et al., 1995).

Besides upper crustal brittle deformations of the LM during its rifting our observations point toward important crustal
reworking at depth accompanied by volcanic extrusions. From the thinned continental crust structure to the west of LM and the lower continental crust velocity signature of the anomalous corridor (Figure 8, but see also MZ1 model in (Leprêtre et al., a)) we infer that oceanward flow of the MCP and NNV lower crust may have 'fed' the corridor. The corridor's crystalline basement is 10-12 km thick on MZ3 (Figure 12c) in the northernmost LM, varies from 8 to 14 km thick offshore the MCP (MZ4 and MZ5 on figure 12c) and thickens to 15-20 km thick east of the NNV (MZ1 on figure 12c). Such variation in thickness may
be evidence of the 'boudinage' of the lower crust originating from its ductile shearing and flow (Clerc et al., 2018; Gernigon et al., 2014; Loureiro et al., 2018). Strong volcanism has accompanied this process as suggested by the 'mixed' composition of the corridor and time constraints given by the deposition on top of these volcanics of the Maputo Sands which preceded the first deep marine sedimentation (Figure 12).

Lower continental crust flow/shearing, eventually leading to exhumation (Aslanian et al., 2009), has been inferred in many
places including offshore eastern Canada and Arctic margins (Gernigon et al., 2014; Reid and Jackson, 1997), along South-Atlantic Brazilian (Evain et al., 2015; Loureiro et al., 2018) and western African margins (Clerc et al., 2018; Moulin et al., 2005), offshore Mozambique (Senkans et al., 2019); and in the Mediterranean Sea Gulf of Lion and Sardinian margins (Afilhado et al., 2015; Moulin et al., 2015; Jolivet et al., 2015). It appears as a common process of rifted continental margin independently of their tectonic context. In fact it has been shown numerically that lower crust may flow across both divergent
margin segments (Huismans and Beaumont, 2011) and strike-slip segments (Le Pourhiet et al., 2017; Reid, 1989). Usually it is noticed that weak lower crust may favor such exhumation process. It was undoubtedly the case for the southern-Mozambique lithosphere which on the verge of its dislocation was affected by a large thermal anomaly (Karoo?) responsible for its anomalous magmatic content. The same anomaly might have favored coeval partial melting and extrusion of volcanic material to form the volcanic basement of the corridor.

**5.3    Rift to drift evolution and vertical movements**

In the alternative East-Gondwana 'fit' proposed by Thompson et al. (2019), the continental domain to the south-west of the Astrid Ridge on the Antarctica plate is the conjugate of the African corner made by the MCP and the Beira High (Figure 151515a-b). Further south the Grunneghona craton is facing the NNV. The exact seaward limit of the Antarctica continental crust is inferred at the location of a strong positive free-air gravity anomaly (Mueller and Jokat, 2019; Scheinert et al., 2016)
since transitional crust is observed further offshore on seismic data (Jokat et al., 2004). At this stage, the Antarctica plate overlaps the corridor which will develop progressively during rifting (Figure 15c) and before oceanic spreading in the MB





starts from Chron M25 ( 155 Ma) onwards (Figure 1515d). As discussed above we infer an intra-continental sag phase for the development of the volcano-sedimentary basin over the MCP and NNV (Figure 21b). Deep crustal magmatic intrusions led to the subsidence of the area without major plate movements ((Figure 15b; Aslanian et al.)). On the other hand rifting along the

LM proceeded from N-S plate motion responsible for the opening of the MB (Figure 15c). The LM developed as a wide shear zone along the eastern margin of the MCP and NNV. The LF acted at this time as a major strike-slip fault while easing the decoupling between upper and lower crust at depth. In the upper crust we infer that strain partitioning is responsible for shear folds, small pull-apart basins and grabens within the volcano-sedimentary basement. Deeper, extension was accommodated by lithospheric flow leading to lower crustal thinning of the eastern fringe of MCP/NNV continental crust and its oceanward

exhumation. This was accompanied by the extrusion of a large amount of volcanics over the corridor. Further west, deep seismic acquisition revealed a transitional domain capped by a thick volcanic layer (the Explora Wedge, e.g. Jokat et al., 2004) off the Antarctic margin which might attest of a equivalent process, thought along a divergent segment. On the African side, extension initially focused in the offshore Zambezi depression before migrating to the south of the Beira High, isolating this continental block (Mahanjane, 2012). The nature of the crust to the north of the Beira High is also deemed to be transitional

with syn-rift magmatism reported (Mueller et al., 2016; Mueller and Jokat, 2019).

According to our stratigraphic analysis, scarce Red Beds continental sedimentation was followed by the widespread deposition of shallow marine Maputo Sands over both the MCP/NNV and LM. Only then, the eastern extremity of the corridor was covered by the first deep marine sediments. Lithospheric and/or asthenospheric flow may have sustained the area to a relative high level during continental rifting (Reid, 1989) allowing continental Red Beds deposits first, then slight subsidence respon-

sible for the first shallow marine incursion. Minor vertical movements must have occurred along the LF only due to strike-slip shearing. It is some times after break-up that larger differential vertical movement took place on either side of the LF between the continental domain and the corridor, hence accentuating the basement high and limiting the westward incursion of the first deep marine sedimentary horizons.

As in previous studies (see Basile, 2015, and references therein) we recognize that a stage of continent-ocean interaction

followed continental rifting and its break-up. Analyzing the possible interplay between the corridor and the mid-oceanic ridge requires, however, a careful analysis of later stratigraphic horizons across and all along the margin which is out of the scope of our study. Figure 15c to 15d illustrates the drift of the mid-ocean ridge axis through time based on identified magnetic chrons (Mueller and Jokat, 2019, and references therein). Valanginian (Figure 15d) corresponds to a period of major kinematic reorganization in the area . While the Patagonian block started its southwestward drift, a triple junction initiated responsible

for enhanced magmatism and the formation of the Mozambique ridge (Fischer et al., 2017). At this time the mid-oceanic ridge axis reached the position of the Ariel Graben. Active deformation thus ceased along the LM which entered in a passive stage and progressively acquired it present morphology.





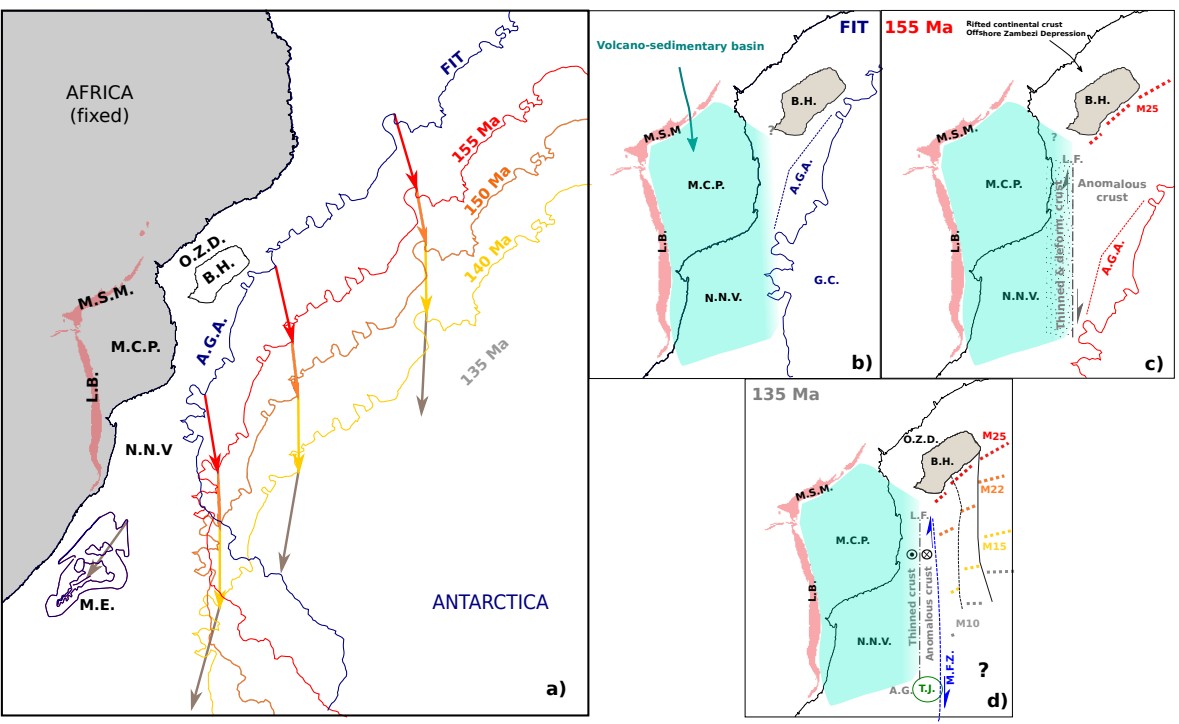

**Figure 15.** Evolution of the Limpopo Margin (LM). a) Kinematic of the Antarctica plate with respect to the African plate fixed according to (Thompson et al., 2019). b,c,d) Zooms on the evolution of the LM with major features annotated respectively before rifting (b), at break-up (c) and after the opening of the Mozambique Basin along the LM (d). Magnetic anomalies are from Mueller and Jokat (2019). LB: Lebombo monocline; MSM: Mateke Sabi Monocline; MCP: Mozambique Coastal Plain; NNV: North Natal Valley; ME: Maurice Ewing bank; BH: Beira high, OZD: Offshore Zambezi Depression AGA: Antarctic Gravity Anomaly; GC: Grunehogna Craton; LF: Limpopo Fault; MFZ: Mozambique Fracture Zone, AG: Ariel Graben. TJ: Triple Junction.

## 6   Conclusions

Within the framework of the PAMELA project we combined plate kinematic reconstruction of East-Gondwana break-up with
seismic constraints on the deep crustal architecture of the southern Mozambique margins. With additional inputs from a tectono-stratigraphic analysis of industrial seismic profiles we infer that strike-slip or highly oblique rifting occurred along the Limpopo segment. There, a wide shear zone formed that is emphasized by a major strike-slip fault zone (LF) in the upper crust and lower crustal flow at depth. We infer that the rifting process led to the thinning of the continental crust beneath the eastern edge of the MCP/NNV and gave the peculiar 'mixed' composition of a 60 km wide corridor observed to the west of the MFZ. Finally, we



show that following break-up decoupling along the LF led to differential vertical motion on each side of the fault with uplift of the continental domain while subsidence affected the corridor.

*Data availability.* Seismic datasets of the PAMELA-MOZ3 (Moulin and Aslanian, 2016) and PAMELA-MOZ5 (Moulin and Evain, 2016) cruises are archived and referenced in Ifremer SISMER database and can be requested at:
https://doi.org/10.17600/16009500 and https://doi.org/10.17600/16001600.

*Author contributions.* The Pamela MOZ35 project was led by M. Moulin, D. Aslanian and M. Evain from Ifremer in collaboration with Total. Modelling of profiles MZ1 and MZ7 was done by A. Lepretre ; MZ2 by F. Verrier and P. Schnurle, MZ3 by M. Evain, profiles MZ4 and MZ5 by L. Watremez and MZ6 by P. Schnürle. Processing of the deep-sounding reflection seismic data was done by P. Schnürle. M. Evain wrote the article with valuable inputs from all co-authors.

*Competing interests.* The authors declare that they have no known competing financial interests or personal relationships that could have
appeared to influence the work reported in this paper.

*Acknowledgements.* We thank the captain, crew, and MCS technical team of the R/V Pourquoi-Pas. We also thank the OBS technical team who maintain and constantly improve our OBS pool, as well as the land stations deployment team. A Leprêtre and F. Verrier respective post-doc studies and contract were co-funded by TOTAL and Ifremer as part of the PAMELA (Passive Margin Exploration Laboratories) scientific project. We thanks WesternGeco, INP and Total for permission to make and publish line-drawings of seismic data. The GMT
(Wessel and Smith, 1998), Seismic Unix (Cohen and Stockwell, 2019; Stockwell, 1999), and Geocluster (CGG-Veritas) software packages were used extensively in this study.





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




**Appendix A**

**Table A1.** Refracted (left) and reflected (rigth) phase names for MZ3 profile, number of explained events, residual mean-squares in s, and normalized chi-squared values

| phase | npts | Trms | $\chi^2$ | phase | npts | Trms | $\chi^2$ |
|-------|------|------|----------|-------|------|------|----------|
| Pw    | 3453 | 0.022 | 0.613 |       |      |      |       |
| Ps1   | 100  | 0.036 | 1.196 | Ps2P  | 681  | 0.037 | 0.160 |
| Ps2   | 818  | 0.029 | 1.198 | Ps3P  | 1237 | 0.033 | 0.195 |
| Ps3   | 1336 | 0.035 | 1.454 | Psv1P | 3037 | 0.037 | 0.194 |
| Psv1  | 1547 | 0.031 | 0.800 | Psv2P | 1836 | 0.038 | 0.350 |
| Psv2  | 2314 | 0.043 | 1.326 | Pg1P  | 2294 | 0.063 | 0.578 |
| Pg1   | 6353 | 0.055 | 1.256 | Pg2P  | 4497 | 0.057 | 0.463 |
| Pg2   | 9062 | 0.049 | 0.467 | Pg3P  | 6140 | 0.061 | 0.326 |
| Pg3   | 1832 | 0.068 | 0.317 | Pg4P  | 5345 | 0.092 | 0.891 |
| Pg4   | 947  | 0.051 | 0.170 | Pg5P  | 4790 | 0.089 | 0.695 |
| Pn1   | 1528 | 0.101 | 0.629 | PmP   | 4296 | 0.082 | 0.444 |
| Pn2   | 157  | 0.144 | 0.899 | Pm2P  | 1303 | 0.130 | 1.034 |

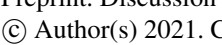



**Table A2.** OBS names and distance along MZ3 model, phase propagation direction, number of explained events, residual mean-square in s, and normalized chi-squared value.

| OBS | shot | dir | npts | Trms | $\chi^2$ | dir | npts | Trms | $\chi^2$ |
|---|---|---|---|---|---|---|---|---|---|
| MZ3OBS01 | 0.000 | -1 | 444 | 0.053 | 1.326 | 1 | 788 | 0.067 | 0.741 |
| MZ3OBS02 | 12.406 | -1 | 561 | 0.101 | 1.592 | 1 | 653 | 0.092 | 3.225 |
| MZ3OBS03 | 24.744 | -1 | 358 | 0.053 | 0.840 | 1 | 563 | 0.059 | 0.405 |
| MZ3OBS04 | 35.074 | -1 | 419 | 0.062 | 0.785 | 1 | 510 | 0.081 | 0.962 |
| MZ3OBS05 | 47.568 | -1 | 480 | 0.081 | 1.570 | 1 | 1117 | 0.067 | 0.382 |
| MZ3OBS06 | 60.108 | -1 | 535 | 0.071 | 0.652 | 1 | 461 | 0.046 | 0.641 |
| MZ3OBS07 | 72.457 | -1 | 914 | 0.079 | 0.878 | 1 | 1571 | 0.063 | 0.231 |
| MZ3OBS08 | 84.855 | -1 | 610 | 0.060 | 0.388 | 1 | 1352 | 0.051 | 0.360 |
| MZ3OBS09 | 97.287 | -1 | 768 | 0.065 | 0.514 | 1 | 1181 | 0.085 | 0.584 |
| MZ3OBS10 | 109.730 | -1 | 578 | 0.043 | 0.402 | 1 | 1268 | 0.075 | 0.830 |
| MZ3OBS11 | 122.245 | -1 | 605 | 0.053 | 0.576 | 1 | 1428 | 0.069 | 0.346 |
| MZ3OBS12 | 134.629 | -1 | 551 | 0.041 | 0.430 | 1 | 1528 | 0.070 | 0.363 |
| MZ3OBS13 | 147.166 | -1 | 1041 | 0.036 | 0.336 | 1 | 1410 | 0.053 | 0.383 |
| MZ3OBS14 | 159.692 | -1 | 845 | 0.044 | 0.419 | 1 | 1032 | 0.049 | 0.362 |
| MZ3OBS15 | 172.186 | -1 | 491 | 0.043 | 0.673 | 1 | 1660 | 0.103 | 0.910 |
| MZ3OBS16 | 186.020 | -1 | 1262 | 0.058 | 0.723 | 1 | 1263 | 0.048 | 0.299 |
| MZ3OBS17 | 197.187 | -1 | 1565 | 0.055 | 0.367 | 1 | 1344 | 0.051 | 0.697 |
| MZ3OBS18 | 209.733 | -1 | 1383 | 0.039 | 0.301 | 1 | 1427 | 0.064 | 0.550 |
| MZ3OBS19 | 222.190 | -1 | 1324 | 0.045 | 0.301 | 1 | 1149 | 0.074 | 0.755 |
| MZ3OBS20 | 234.697 | -1 | 1430 | 0.059 | 0.440 | 1 | 1023 | 0.076 | 0.568 |
| MZ3OBS21 | 247.110 | -1 | 1481 | 0.079 | 0.701 | 1 | 1124 | 0.054 | 0.459 |
| MZ3OBS22 | 259.541 | -1 | 1803 | 0.057 | 0.370 | 1 | 872 | 0.057 | 0.446 |
| MZ3OBS23 | 271.942 | -1 | 1699 | 0.062 | 0.454 | 1 | 766 | 0.061 | 1.460 |
| MZ3OBS24 | 284.437 | -1 | 1656 | 0.053 | 0.323 | 1 | 812 | 0.045 | 0.474 |
| MZ3OBS25 | 296.936 | -1 | 1722 | 0.076 | 0.925 | 1 | 767 | 0.059 | 0.938 |
| MZ3OBS26 | 309.420 | -1 | 1692 | 0.067 | 0.778 | 1 | 660 | 0.073 | 1.335 |
| MZ3OBS27 | 321.914 | -1 | 958 | 0.079 | 0.691 | 1 | 989 | 0.063 | 0.660 |
| MZ3OBS28 | 334.449 | -1 | 915 | 0.109 | 1.204 | 1 | 1083 | 0.074 | 0.484 |
| MZ3OBS29 | 346.948 | -1 | 781 | 0.115 | 1.727 | 1 | 1211 | 0.068 | 0.885 |
| MZ3OBS30 | 359.463 | -1 | 658 | 0.040 | 0.654 | 1 | 1078 | 0.048 | 0.575 |
| MZ3OBS31 | 372.017 | -1 | 1028 | 0.081 | 0.920 | 1 | 883 | 0.061 | 1.136 |
| MZ3OBS32 | 384.520 | -1 | 943 | 0.062 | 1.108 | 1 | 813 | 0.059 | 0.656 |



## Appendix B: MZ3 uncertainties estimation

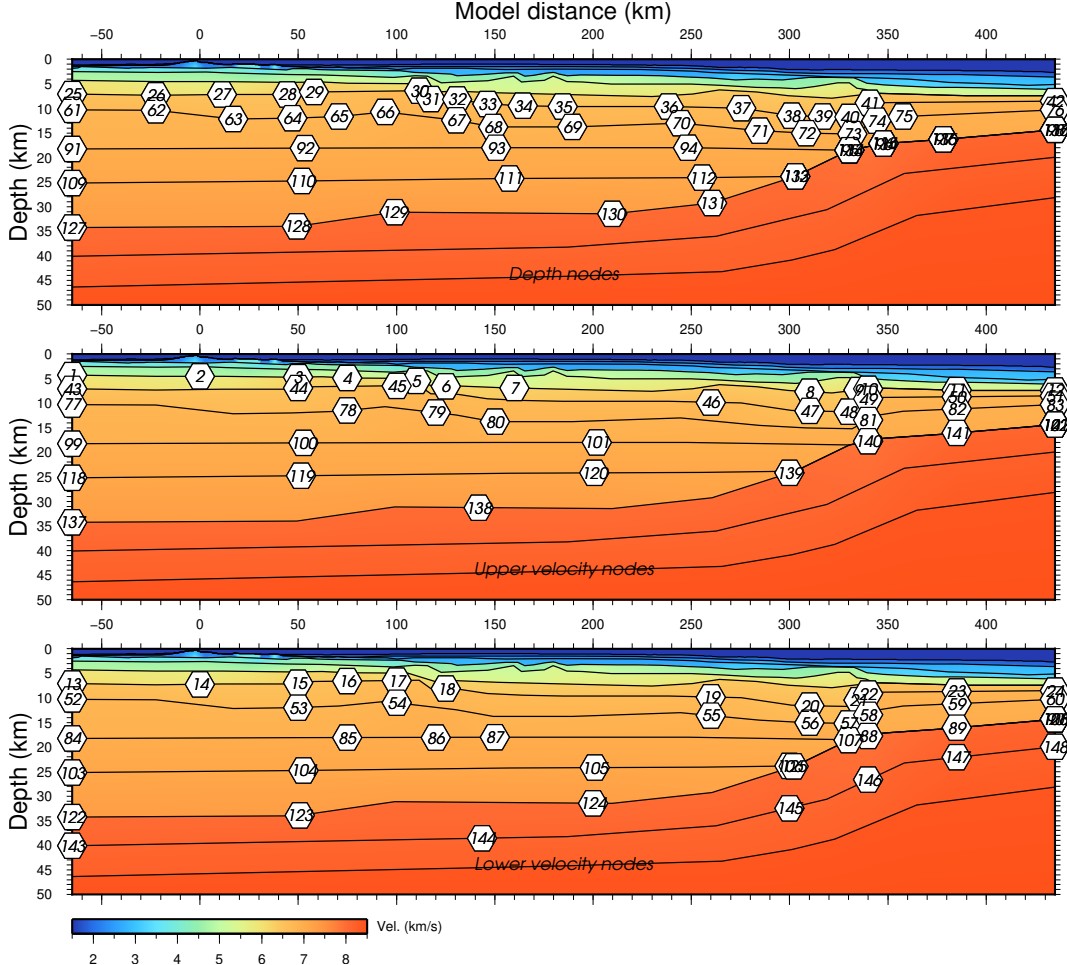

**Figure B1.** MZ3 final velocity model with parameter selection (white hexagons) for Vmontecarlo. Top panels: depth parameters; middle panels: top of layer velocity parameters; and bottom panels: base of layer velocity parameters. Interfaces indicated by black lines and velocities colored according to color.





## Appendix C: MZ3 PSDM

**Figure C1.** Pre-stack depth migration of MCS MZ3 profile. Left: SW portion ; right: NE portion. a) PSDM record section. The intersections with the MOZ35 dataset are indicated by red line. Vertical exaggeration is 1:10. b) Residual move-out. c) Semblance plot of the RMO. Common image gathers are spaced every 7.5 km. Vertical exaggeration is 1:15. Model's interfaces are represented with continuous lines.