# Peer review of "Crustal structure of the East-African Limpopo Margin, a strike-slip rifted corridor along the continental Mozambique Coastal Plain and North-Natal Valley"

_Solid Earth, 2020_

## Author Response (AR1)

**RC1**: 'Comment on se-2020-209', Anonymous Referee #1, 16 Mar 2021 reply

This manuscript presents new images of the crust and uppermost mantle along a profile offshore the Mozambique Coastal Plain. The authors analyze multichannel and wide-angle seismic data using mature software (CGG-Veritas Geocluster and RAYINVR). The P-wave tomography results show a thinned crust in the northeastern end of the Limpopo margin, in contrast to the 34-km-thick crust in the southwestern part of the Limpopo margin and the North Natal Valley. The authors calibrate their tectonostratigraphic analysis with industrial well logs, and suggest a strike-slip rifting process around 155 Ma ago that separated Antarctica and Africa plates.

I cannot make comments on data analysis as that is beyond my specialty. Although the discussions are thorough, I still find it hard to follow as a reader who is not familiar with this region. Low-quality figures prevent me from evaluating the results in depth. Here are some suggestions on presenting.

- The abstract is too long. It is challenging to get the take-away message from the abstract.
- The font sizes in almost all figures are too small, making it difficult to evaluate the results and understand the discussions.
- It would be better to show a large-scale tectonic map or plate reconstruction map as the first figure to explain the East-Gondwana break-up discussed in the introduction section.
- I understand Fig. 8 is the primary product of data analysis. How is it related to the following discussions? It would be helpful to label certain features on top of the tomography image.
- All geographic features mentioned in this paper need to be labeled in Fig. 1.

**Citation**: https://doi.org/10.5194/se-2020-209-RC1

Reply to RC 1

We first would like to thank Referee #1 for his time to review our manuscript despite being unfamiliar with the study area and this type of data analysis. We understand that the initial version of our manuscript might have been hard to read. Therefore, we have prepared a new version that will hopefully be easier to read and follow. We made a full review of our manuscript to improve its clarity and respond to the community and the two referee comments. It now includes, as suggested by Referee #1:
- a shorter and more straightforward abstract,
- updated figures with improved visibility,
- an introductory figure that pictures the general geodynamic framework of the studied region,
More generally, we paid particular attention that all geographic locations and features mentioned in the text are visible on figures. We also extended figure 1 (which is now figure 2) which should now summarizes most of the cited geographical locations. With respect to figure 8 (now figure 12), which is indeed the primary product of our study, it has been moved further down the text in the section 5 which specifically focus on our results and their interpretations. Within this section we systematically refer to this figure (the velocity model) to present a top to bottom view of the crustal structure of the margin that is key for the following discussion.

Best regards,
Mikael Evain on behalf of all co-authors

**RC2**: 'Comment on se-2020-209', Anonymous Referee #2, 28 Mar 2021

This manuscript presents results from a recent active source seismic survey offshore the coastal plain of Mozambique. These results are interpreted in terms of the past breakup of Gondwana, and kinematic plate reconstructions are explored to explain the results. Unfortunately, in its current state, I had difficulty in following what the main new findings were from the geophysical analysis and how they supported the conclusions.

As a reader who was unfamiliar with the detailed tectonic history of the region, I found it difficult to follow the main points presented in this manuscript. There are references to many local geographic and geologic features throughout, as well as seismic profiles, which is fine. But it would be helpful if all of the major features that are discussed in the manuscript are introduced on one of the earlier figures.

This may also in part be due to my unfamiliarity with the particular region, but I found it very difficult to follow how the new results presented in this manuscript supported the conclusions related to the tectonics and geodynamics of the region. Perhaps a summary of the main, new structural findings from the seismic and gravity analyses that are then referred to throughout the discussion would help. Additionally, summary figures with the major interpreted features in the geophysical results labeled may also help.

The size of the figure text significantly affected my ability to understand the manuscript easily. Some of the figures, or parts of the figures showing the seismic results may be able to be moved to the supplement.

There are minor grammar mistakes throughout, as well as occasional other errors suggesting a level of incompleteness such as incomplete citation information, underscores in figure axes labels, etc. These should be cleaned up before publication.

**Citation**: https://doi.org/10.5194/se-2020-209-RC2

Reply to RC2

We would like also to thank Referee #2 for his time to review our manuscript despite the difficulties faced. As mentioned in our reply to Referee #1, we have prepared a new version that will hopefully be easier to read and follow. We fully reviewed our manuscript to improve its clarity and respond to the community and the two referee comments. Among others we can mention that:
-all the figures have been improved for visibility and we paid particular attention they include all geographic localities mentioned in the text.
-we fully reworked our abstract for a more straightforward message
-we clarify our introduction which now better replaces this study within the scope of the larger Pamela Moz3-5 project. We further added an introductory figure presenting a broader geodynamic picture of south-eastern Gondwana breakup
-section 2 on geological settings now clearly mentions existing controversies about the area and further detail geological arguments that support them. Most specifically it includes dedicated paragraphs on the crustal nature of the MCP/NNV domain, the age of the Mozambique basin's oceanic crust and controversies regarding the Mozambique ridge.
-sections 3 and 4 now exclusively concentrate on data analysis, seismic modeling and validation of our final velocity model
-section 5 present all our results and our interpretation of the crustal structure of the Limpopo margin. It summarizes our findings before they are discussed in sections 6. We truly hope this will

clarify the general message that our manuscript try to convey and how it is supported by geophysical evidences.

Best regards,
Mikael Evain on behalf of all co-authors

---

## Author Response (AR2)

**Topical Editor Decision: Publish subject to minor revisions (review by editor)** (24 Jun 2021) by
Caroline Beghein
Comments to the Author:
Dear authors,

After receiving comments back from one of the original reviewers, I have decided to recommend publication provided you make minor revisions to the figures. I agree with the reviewer that many of them are difficult to read due to the small font and the paper would benefit from increasing the font on some figures (Figs. 2, 4 to 9, 13, 16, C1 and Supp Fig 3) and perhaps splitting other figures in two if readability is not improved by increasing font (e.g. Fig. 4, 13).

Dr. Caroline Beghein, PhD

########################################################################

Reply to Topical Editor

Dear  Dr. Caroline Beghein,

On behalf of all co-authors of this manuscript I would like to thanks you for handling the revision process. I would like also to thank the editorial team and the reviewers for their time and work.
I have carefully reviewed all the figures of the manuscript and made the necessary adjustments in font size for them to be readable at the scale of plain page. I separated in two Figure 4 as suggested. I found it difficult to do the same for Figure 13 but rotated it instead.

Thanks again to Solid Earth for accepting our manuscript. I am looking forward to its final publication.

Best regards,
Mikael Evain & co-authors